# Mineralogical controls on PFAS and anthropogenic anions in subsurface soils and aquifers

Marina G. Evich [1] ✉, James Ferreira[2], Oluwaseun Adeyemi[3], Paul A. Schroeder [3], Jason C. Williams[4], Brad Acrey[5], Diana Burdette[5], Malcolm Grieve[5], Michael P. Neill[5], Kevin Simmons[5], Brian C. Striggow[5], Samuel B. Cohen[1], Mike Cyterski[1], Donna A. Glinski [1], W. Matthew Henderson [1], Du Yung Kim[1] & John W. Washington [1,3] ✉

Per- and polyfluoroalkyl substances (PFAS) migrate into the environment through various means, e.g., soil-amendment impurities and ambient atmospheric deposition, potentially resulting in vegetative uptake and migration to groundwater. Existing approaches for modeling sorption of PFAS commonly treat soil as an undifferentiated homogeneous medium, with distribution constants (e.g., $K_d$, $K_{oc}$) generated empirically using surface soils. Considering the limited mineral variety expected in weathered geologic media, PFAS mobility can be better understood by accounting for predictable mineral assemblages that are ubiquitously distributed in US soils. Here we explore the role of minerals and electrostatic sorption in controlling PFAS mobility in subsurface settings at contaminated agricultural sites by measuring geochemical parameters and PFAS, and calculating pH-dependent mineral surface charges through full soil and aquifer columns. These data suggest subsurface mobility of short-chain PFAS largely is controlled by aluminum-oxide mineral(oid) electrostatic sorption, whereas long-chain PFAS mobility is controlled by organic matter and air-water interfacial area.

Our terrestrial environment is ubiquitously impacted by a wide array of anthropogenic compounds, the majority of which have been produced for their bioactive or other functional properties. Remote from their intended use, in the environment, these chemicals maintain the potential to exert effects on their surroundings, potentially including humans, fauna, flora, fungi, and microbiomes. To sustain our way of life, insecticides, acaricides, herbicides, and fungicides are applied liberally to agricultural crops and soils. Livestock are fed antibiotics, steroids and other agents, and their waste and other solid wastes are spread on agricultural fields. In recent years, more than half of the biosolids generated in municipal wastewater-treatment plants in the US were applied as amendments to agricultural and other soils[1]. Often these biosolids bear a witches' brew of our societal waste, e.g., pharmaceuticals, PFAS, and plasticizers[2]. Still more anthropogenic compounds have been documented to undergo long-range atmospheric transport from air emissions at manufacturing facilities[3,4], as well as transport and deposition of contaminant coated aerosols during sand and dust storms[5]. Released to the land's surface, these pesticides, and manure-, biosolids- and atmospheric-borne chemicals interact intimately with soils, which initially act as sinks from the aqueous phase, serving as sizable reservoirs, and ultimately time-delayed sources for moderated concentrations of recalcitrant contaminants, perhaps to be

[1]USEPA, Office of Research and Development, Center for Environmental Measurement and Modeling, Athens, GA, USA. [2]USEPA, Region 4, Atlanta, GA, USA. [3]Department of Geology, University of Georgia, Athens, GA, USA. [4]South Carolina Department of Health and Environmental Control, Bureau of Land and Waste Management, Columbia, SC, USA. [5]USEPA, Region 4, Laboratory Services and Applied Sciences Division, Athens, GA, USA. ✉e-mail: evich.marina@epa.gov; washington.john@epa.gov

accumulated in vegetation or soil organisms[6] or to leach through the vadose zone and migrating with groundwater[7,8]. Critically, as groundwater supplies roughly half of potable drinking water globally[9], these fugitive solutes present a considerable potential route of exposure to global populations.

Prominent among organic contaminants are per- and polyfluoroalkyl substances (PFAS), which have been found in biosolids[10] and biosolids-amended agricultural fields[11,12]. PFAS, an extensive family of anthropogenic compounds found in both industrial and consumer products[13], usually are composed of fluorinated aliphatic chains terminating in relatively hydrophilic functional head groups[14], imparting high resistance to thermal, chemical, and microbial degradation[13], and leading to environmental persistence as well as substantial challenges in modeling transport or large-scale remediation efforts[13]. Among PFAS uses, the widespread industries of textile[15], paper[16], and carpet[17] milling operations historically have used polymeric PFAS, e.g., sidechain fluorotelomer polymers, to treat their products for anti-staining properties[18]. These polymers slowly degrade by abiotic hydrolysis, with half-lives on the order of 55 to 90 years under environmental conditions, to form fluorotelomer alcohols (n:2FTOHs−F$(CF_2)_n(CH_2)_2$OH, $n$ ‑ 4–18 even integers) as first-generation products[19], and ultimately forming recalcitrant perfluorocarboxylates (PFCAs−F$(CF_2)_n$CO(OH), $n$ ‑ 3–18) and related compounds in the environment by microbially mediated transformations[19,20]. Along with recalcitrant perfluorosulfonates (PFSAs−F$(CF_2)_n$SO$_3$H, $n$ ‑ 4–12), PFCAs are among the most widely distributed PFAS in the environment[13].

Despite this considerable potential for PFAS and other contaminants to migrate from soil to agricultural and water resources, currently contaminant subsurface mobility often is modeled by treating soils as homogeneous and isotropic bodies with simple approaches such as single distribution coefficients derived from organics-rich surface soils[13], e.g., $K_d$ or $K_{oc}$ values, to represent the entire soil column. However, the subsurface is composed almost entirely of minerals, and it is well established in geochemistry that soil bodies vary considerably and predictably with depth in mineralogic assemblage, chemical and physical properties[21]. While the role of minerals in the subsurface fate of ionized contaminants (including PFAS) is appreciated conceptually[22], the relative importance of specific minerals and assessment of their properties affecting contaminant fate in real-world soil bodies, are less well explored.

To better understand the complex processes involved in PFAS fate in the terrestrial subsurface, we investigated two sites that received sludge applications. The sludge originated from a now-defunct textile mill, the former Galey & Lord facility in South Carolina, which was designated an EPA Superfund site in 2022 (EPA ID SCD058189622). At each study site, roughly 50 soil-core segments were collected from the surface, through the vadose zone and water-table aquifer, to an aquiclude at 18 m, and analyzed for PFAS and an exhaustive suite of geochemical and mineralogic properties (see Supplementary Information section SI I Supplementary Methods). Here we report PFAS distribution patterns through the subsurface, document the ubiquitous distribution of authigenic minerals in soils across the United States, and explore how these globally distributed minerals and other geologic soil parameters control distribution and fate of PFAS in the subsurface (SI II Data Summary).

## Results
### Geochemistry of mineral weathering
Soils, sediments, and weathered-aquifer zones form from igneous, metamorphic, and sedimentary facies ranging in normative elemental compositions from granitic (alumino-silicic rich) to basaltic (ferro-magnesian rich) mineral assemblages[23]. Most minerals that are thermodynamically or kinetically favored during rock formation are not stable when subjected to weathering, undergoing dissolution or

recrystallization at varying rates[24]. While some of these minerals are resistate, reacting slowly to weathering, e.g., commonly occurring quartz (SiO$_2$), most weather at rates that are fast relative to soil-forming processes (SI III Extended Geochemical Summary). As weathering of primary rock minerals proceeds, a series of a few authigenic mineral families form by precipitation, then with continued weathering may dissolve to form others, so that weathered settings can be classified by mineral assemblages (Fig. 1). These weathering regimes have been classified as incipient, intermediate, advanced, and extreme[25] (Fig. 2A), and their distribution in the United States is exhaustively mapped by United States Department of Agriculture (USDA) Soil Order, e.g., inceptisols, alfisols, ultisols and oxisols, among others[26] (Fig. 2A). In stable soil profiles, these weathering regimes commonly vary with depth, incipient at the bedrock interface, grading to intermediate in B horizons, and perhaps advanced in the uppermost A and E horizons. Over decades of effort, the USDA has accumulated an extensive soil database[27] including the content of these mineral phases in US soils, documenting nearly total ubiquity of authigenic minerals across all Soil Orders (e.g., Table 1 and Supplementary Tables 19–21) and trends in assemblage with increasing weathering (Supplementary Fig. 6).

Based upon geochemical principles (SI III Extended Geochemical Summary) and as documented in the USDA database[27], incipient and intermediate weathering regimes commonly are enriched in clays such as illite, vermiculite, smectites (Fig. 1 and Table 1) which bear a permanent negative surface charge offering the potential for electrostatic sorption of cationic compounds and/or anionic compounds with cation bridging. Advanced and extreme weathering regimes commonly are enriched in kaolinitic clays and (hydr)oxide minerals, e.g., goethite, hematite, gibbsite[24,25] (Fig. 1 and Table 1), bearing pH-dependent, variable surface charges that commonly are positive under soil conditions (SI III Extended Geochemical Summary), thereby offering potential sorption substrates for anionic compounds.

Critically with regard to anionic-compound mobility, crystalline authigenic minerals generally do not nucleate directly from aqueous soil solutions poised at the cusp of mineral saturation. Instead, because of high nucleation barriers for thermodynamically stable mineral phases, metastable amorphous phases (mineraloids) nucleate first, heterogeneously on existing surfaces[28], inducing precipitation of the crystalline mineral phases in a process known as Ostwald ripening[28]. Prominent among these amorphous phases, hydrous aluminum oxide (HAO; roughly Al(OH)$_3$) and hydrous ferric oxide (HFO, roughly Fe(OH)$_3$), have particularly high surface areas (Supplementary Table 5) with pH- and ionic-strength- dependent electrostatic surface charges that offer rich sorption complexes for charged compounds[29,30]. These metastable amorphous mineraloids are nearly ubiquitously distributed in US soils (Table 1 and Supplementary Tables 19–21) and, by inference, elsewhere globally.

### Surface and subsurface PFAS distribution
In terrestrial settings, PFAS partition between the aqueous phase and solid surfaces, including NOM that is concentrated in surface soils and organic-rich sediments[31]. Bearing hydrophobic chains, in vadose settings, PFAS also have been shown to sorb to the air-water interface[32]. Some fraction of PFAS binding to these interfaces is reversibly sorbed, and PFAS uptake into vegetation, including agricultural food products[33], and/or percolation to aquifers[7] has been widely reported[34]. However, in subsurface settings, where NOM is generally much less concentrated than in surface soils, details regarding sorption and release of PFAS, which govern these uptake and percolation processes, are ill-defined.

We studied PFAS distribution through full soil columns at a PFAS impacted colluvial-soil site (Field 1) and residual-soil site (Field 2), characterizing subsurface vadose and aquifer samples for PFAS and a full range of geochemical/mineralogical properties

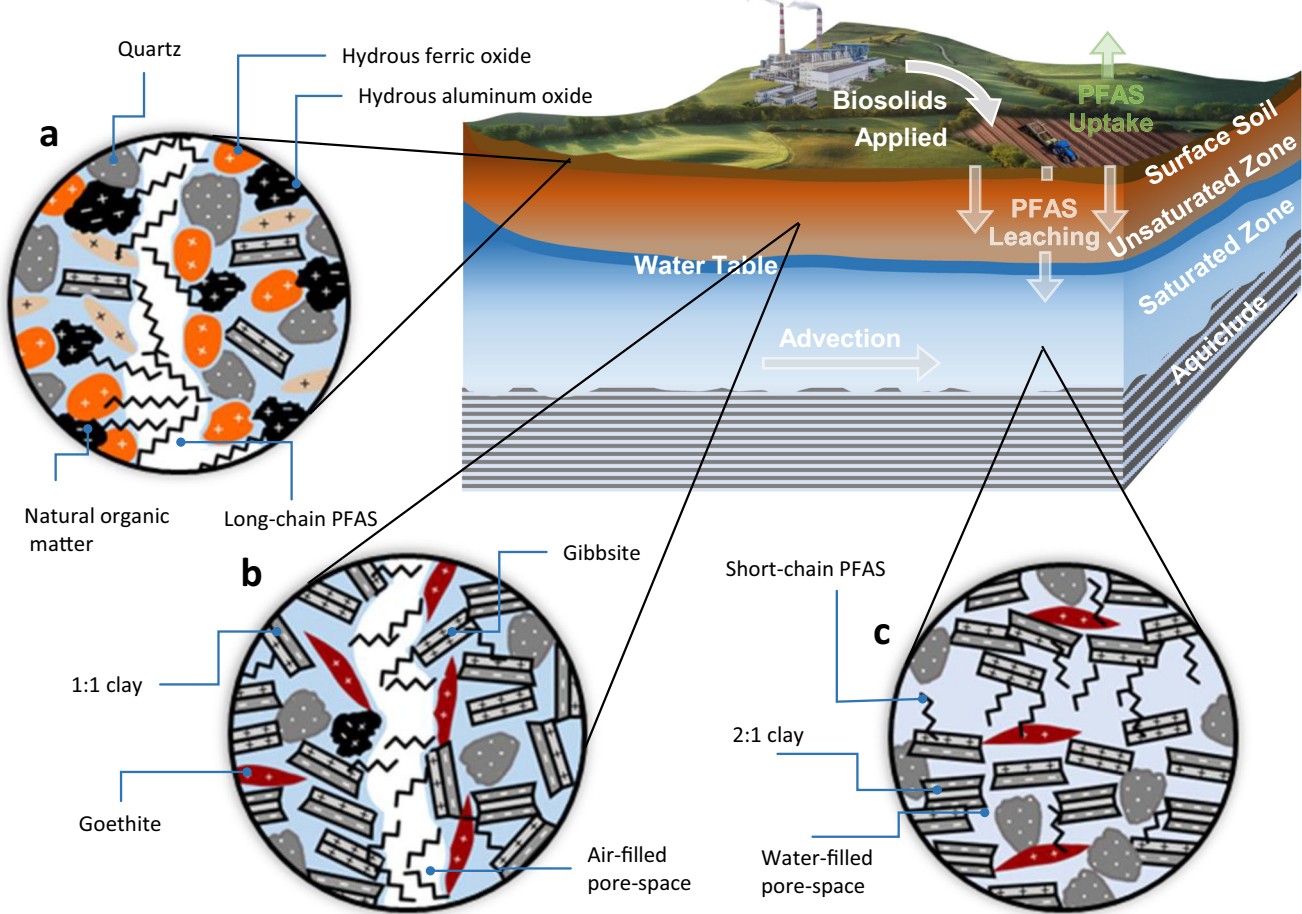

**Fig. 1 | Anthropogenic compounds applied to soils partition between soil solids, water and air-filled pore space. a** Surface soils concentrate in weathering-resistant quartz, organic matter, and low concentrations of incipient amorphous Fe and Al (hydr)-oxides where long-chain PFAS are retained. **b** Subsurface horizons accumulate authigenic minerals, including clays and crystalline Fe and Al (hydr-) oxides where short-chain PFAS can sorb. **c** In saturated media, mineral assemblage commonly reflects more extensive weathering in high-flow zones relative to low-flow zones. Al (hydr)oxides and kaolinite can sorb short-chain PFAS regardless of oxidation state, while in reduced settings Fe (hydr)oxides may be subject to reductive dissolution thereby not contributing to the potential exchange complex.

(SI I Supplementary Methods). At our study sites, PFCAs comprised the dominant class of PFAS (SI II Data Summary), 86% and 93% of total target PFAS in surface soils in Fields 1 and 2 respectively (Supplementary Tables 14 and 15), consistent with expectations for soils having received textile-based sludge applications. Wastewater biosolids and sludges, including urban (e.g., New York City) and industrially impacted, have been widely documented to be high in PFAS[35]. Among sludge sources, sludges originating from textile operations can be particularly high in PFAS[15,36], largely as a consequence of textiles commonly having been treated for water-repellency using sidechain fluorotelomer polymers[37,38], which have been documented to degrade to PFAS in the environment[19,20] as described above.

Following at least 16 years since the most recent application of sludge (SI I, Supplementary Fig. 1), PFCAs are still most concentrated in the surface soil compared to subsurface soil, with [PFCAs] ~ 225 μg/kg for colluvial Field 1 (dry-soil mass; Fig. 3A and Supplementary Fig. 14). Below the surface soil, subsurface maxima are present in both study sites at roughly 1–2 m depth (Fig. 3A and Supplementary Fig. 20), a depth interval typical for B horizon soils where authigenic soil minerals often are most concentrated (Table 1 and Supplementary Tables 19–21), but these subsurface PFAS maxima are roughly sixfold lower than at the surface (e.g., for colluvial Field 1 [∑PFCAs]$_{1.67\,m}$ = 34.1 μg/kg).

In terms of distribution among individual PFCAs, the surface soils are dominated by the longer chain compounds (Fig. 3B), with perfluorodecanoic acid [PFDA, C10 (total carbon count, including the carboxylate carbon)] through perfluorotetradecanoic acid (PFTeA, C14) comprising >90% of PFCAs. In contrast, for the subsurface, shorter chain PFCAs dominate (Fig. 3B) with perfluorobutanoic acid (PFBA, C4) through perfluorooctanoic acid (PFOA, C8) comprising >90% of PFCAs. At the nexus between the poorly mobile long-chain PFCAs (C10–C14) and relatively more mobile short-chain PFCAs (C4–C8), perfluorononanoic acid (PFNA, C9, bearing $(CF_2)_8$) exhibits a distribution intermediate between these compounds, with detectable concentrations throughout the soil column, albeit at lower fractions than its shorter homologues (Fig. 3B and Supplementary Fig. 22).

With sludge from a textile mill as the source of these PFAS, the historically used sidechain fluorotelomer polymers are inferred to be the dominant source of the PFCAs[18,38]. The n:2FTOHs bound to these polymers bear even numbers $n$ of $(CF_2)_n$ groups, typically on the order 3% $n$ = 6, 50% $n$ = 8, 30% $n$ = 10, 10% $n$ = 12 and 3% $n$ = 14[20]. Under environmental conditions, these FTOHs are subject to β- and lesser amounts of α-oxidation to form the $(CF_2)_n$ = (n-1) and $(n)$ PFCAs, for example 8:2FTOH → PFOA (C8; n-1) + PFNA (C9; $n$)[20,39]. Accordingly, summing the observed PFCAs at our study site by this formula, compared to typical sidechain fluorotelomer polymers, surface soils are richer in long-chain PFCAs $(CF_2)_n$ $n$ = 9–13 (Fig. 3C) and subsurface soils are richer in short-chain PFCAs $(CF_2)_n$ $n$ = 3–7 (Fig. 3D), reflecting preferentially high mobility of short-chains compared to long-chains.

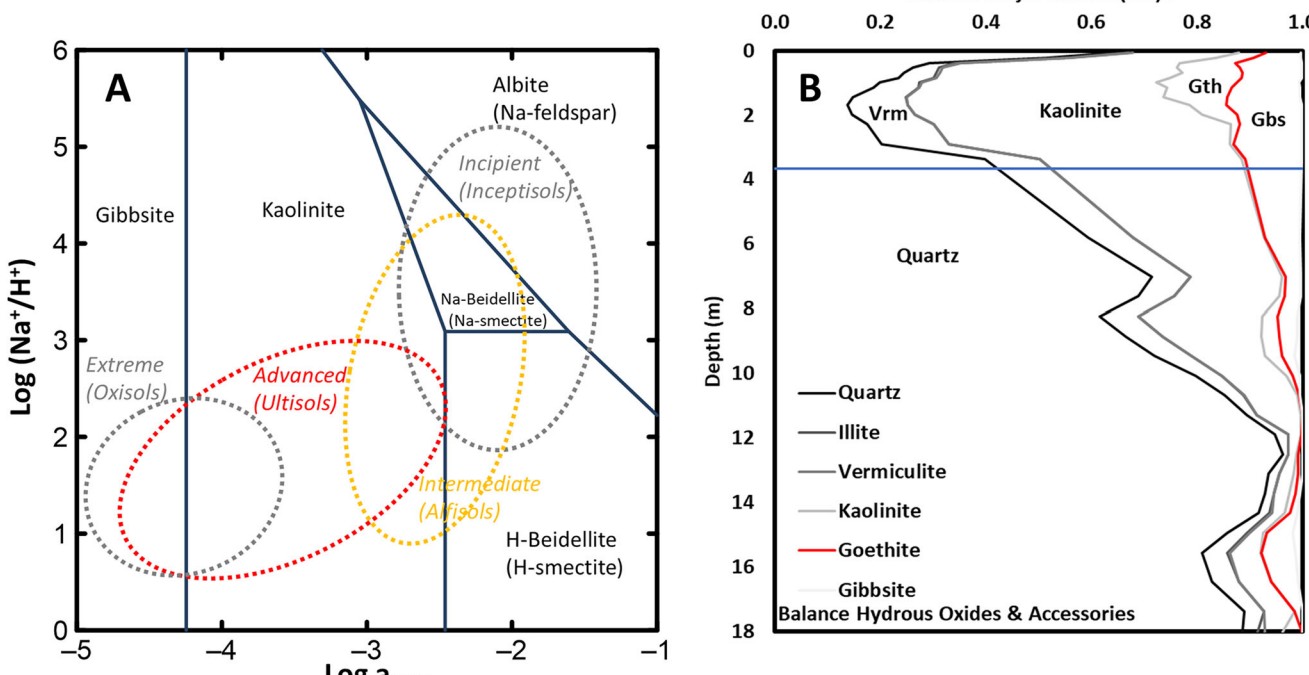

**Fig. 2 | Major minerals expected to form in soils from weathering and found at the present study sites. A** Weathering dissolution of a wide array of rock minerals produces a limited assemblage of authigenic minerals that are present in almost all soils and weathered settings, the abundance of which evolves with degree of weathering. Here thermodynamically stable minerals[61] are plotted as a function of evolving soil-water composition, with general weathering classifications and USDA Soil Orders depicted. In this example of the global soil-forming process, the

common rock-forming mineral albite dissolves, first tending to form 2:1 clays (two silica sheets sandwiching an aluminum sheet). As weathering continues, 2:1 clays dissolve to form 1:1 clays and, in still more weathered zones, the 1:1 clays dissolve leaving (hydr)oxide minerals. **B** Mineralogic profiles using data from the current study (3-point smoothed), dominated by resistate quartz and authigenic kaolinite and gibbsite, with some vermiculite (a 2:1 clay) as expected for soils of advanced weathering stage. The blue horizontal line depicts the water table.

However, when PFCAs are summed over the entire soil column, weighted by sampling increment, the homologue profile for the poorly mobile long-chains $(CF_2)_n$ $n = 10$–$14$ compares quite closely with the sidechain fluorotelomer polymeric source (Fig. 3E). The homologue distribution for the whole soil column is slightly depleted in PFOA + PFNA compared to the sidechain fluorotelomer polymer (Fig. 3E), reflecting PFOA's relatively higher mobility (Fig. 3B). The whole soil column also sums to higher short chains than the historic longer sidechain fluorotelomer polymers (Fig. 3E), perhaps reflecting minor transformation reactions that can generate shorter chain PFCAs[13,20] or use of newgeneration sidechain fluorotelomer polymers bearing shorter chains[40].

Following PFCAs, PFSAs were the second-most dominant family of PFAS in Field 1 (Supplementary Tables 14 and 15) but not Field 2. Field 1 received textile sludge more recently than Field 2 (Supplementary Fig. 1), likely when alternative textile treatments were in use. PFSAs are not unexpected in textile waste as some sidechain polymers used in textile industry use sulfonamido-ethanol sidechains which can degrade to PFSAs[18,38]. Bolstering this premise, intermediate compounds including N-ethyl perfluorooctane sulfonamidoacetic acid[13] were detected in the surface soils (Supplementary Table 14). In contrast to PFCAs, the largest part of PFSAs had chain lengths of $(CF_2) \leq 8$. In the absence of longer chain lengths, the longest PFSA detected in Fields 1 and 2 are $(CF_2)_n$ $n = 10$ and $n = 8$, the ∑PFSAs were highest in the shallow subsurface (<3 m), generally at similar depth increments to the PFCA subsurface maxima at the B-horizon depths (Supplementary Fig. 21). Additionally, the PFSAs showed similar relative mobility patterns to the PFCAs among homologue lengths; for example, perfluorobutane sulfonate (PFBS, C4) maximum fell at 1.5 to 3 m, whilst perfluorooctane sulfonate (PFOS, C8) maximum was at <1 m (Supplementary Fig. 21).

The patterns observed in relative mobilities amongst homologues of PFCAs and PFSAs also are present in the groundwater, with short-chains detected (C4-C10 for PFCAs, C5 to C9 for PFSAs), but longer homologues remaining at non-detectable aqueous concentrations (Fig. 3F). In terms of potential exposure from ingestion of groundwater, the observed concentrations of the short-chain PFCAs and PFSAs are considerably higher than EPA's 2016 and superseding interim 2022 health advisories[41,42] (Fig. 3G). The combined factors of (i) the detection of short-lived precursors observed 16 years since the last sludge application, (ii) the persisting close comparison of the soil-column homologue distribution with sidechain fluorotelomer polymers used to treat textiles, and (iii) the groundwater short chain PFAS concentrations exceed drinking-water health advisories by orders of magnitude, in concert suggest that slow degradation of sidechain fluorotelomer polymers may contribute PFAS to soil and groundwater for decades or longer into the future, consistent with our previous laboratory experiments and modeling[19,20].

We employed the generalized two-layer model of Dzombak and colleagues[29,43] to calculate (Supplementary Eqs. 1–16) the pH-dependent electrostatic charge (Fig. 4A, B) on the aluminum mineral(oid)s hydrous aluminum oxide (HAO), gibbsite and kaolinite, and the ferric mineral(oid)s hydrous ferric oxide (HFO) and goethite (SI I Supplementary Methods). As expected, these phases present considerable positive surface charges through the soil column, offering a rich potential sorption matrix for the anionic PFAS present in our study and which are the PFAS most commonly encountered in contaminated environments[13]. Consistent with geochemical principles (SI III Extended Geochemical Summary) and the extensive USDA database of US soils (Table 1 and Supplementary Tables 19–21), the amorphous mineraloids' HAO and HFO charge contributions are most concentrated at the soil-column surface, and the crystalline minerals

**Table 1 | Summary of B horizon mineral(oid) content for soils in USDA database**

| B Horizon Mineral(oid) | Statistic (count) | Increasing Pedogenesis | | | | |
|---|---|---|---|---|---|---|
| | | Entisol | Inceptisol | Alfisol | Ultisol | Oxisol |
| **Semi-quantitative X-ray proportion of clay-size fraction (1)** | | | | | | |
| **Quartz** | $\bar{X} \pm \sigma$ (n) | 1.2 ± 0.5 (131) | 1.1 ± 0.4 (612) | 1.2 ± 0.4 (2085) | 1.2 ± 0.6 (494) | 2.1 ± 1.3 (22) |
| | Median | 1.0 | 1.0 | 1.0 | 1.0 | 4.0 |
| **Vermiculite** | $\bar{X} \pm \sigma$ (n) | 1.9 ± 0.8 (9) | 1.9 ± 0.9 (160) | 2.0 ± 0.8 (354) | 1.7 ± 0.7 (102) | 1.7 ± 0.6 (3) |
| | Median | 2.0 | 2.0 | 2.0 | 1.0 | 2.0 |
| **Montmorillonite** | $\bar{X} \pm \sigma$ (n) | 2.6 ± 1.1 (256) | 2.6 ± 1.1 (572) | 2.8 ± 1.0 (2749) | 2.0 ± 1.0 (228) | 1.1 ± 0.4 (7) |
| | Median | 3.0 | 3.0 | 3.0 | 2.0 | 1.0 |
| **Kaolinite** | $\bar{X} \pm \sigma$ (n) | 2.2 ± 1.0 (320) | 2.3 ± 1.0 (1445) | 2.7 ± 0.8 (3894) | 3.3 ± 1.0 (1503) | 3.3 ± 1.0 (264) |
| | Median | 2.0 | 2.0 | 3.0 | 3.0 | 3.0 |
| **Goethite** | $\bar{X} \pm \sigma$ (n) | 1.0 ± 0.2 (24) | 1.3 ± 0.5 (266) | 1.4 ± 0.5 (893) | 1.4 ± 0.5 (956) | 1.7 ± 0.6 (282) |
| | Median | 1.0 | 1.0 | 1.0 | 1.0 | 2.0 |
| **Hematite** | $\bar{X} \pm \sigma$ (n) | 1.2 ± 0.4 (6) | 1.2 ± 0.4 (110) | 1.1 ± 0.3 (422) | 1.3 ± 0.5 (475) | 1.7 ± 0.6 (146) |
| | Median | 1.0 | 1.0 | 1.0 | 1.0 | 2.0 |
| **Gibbsite** | $\bar{X} \pm \sigma$ (n) | 1.7 ± 0.9 (15) | 1.8 ± 0.7 (283) | 1.5 ± 0.7 (135) | 2.0 ± 0.8 (571) | 1.9 ± 0.8 (187) |
| | Median | 2.0 | 2.0 | 1.0 | 2.0 | 2.0 |
| **Quantitative percent (geometric mean & standard deviation) of dry soil mass** | | | | | | |
| **HFO** | $\bar{X}_g$ (n) | 0.13 (184) | 0.37 (1758) | 0.24 (1664) | 0.19 (1171) | 0.25 (248) |
| | $\sigma_g$ range | 0.05 - 0.37 | 0.13 - 1.08 | 0.09 - 0.67 | 0.06 - 0.62 | 0.09 - 0.72 |
| | Median | 0.14 | 0.40 | 0.27 | 0.21 | 0.26 |
| **HAO** | $\bar{X}_g$ (n) | 0.12 (189) | 0.24 (1769) | 0.13 (1664) | 0.18 (1174) | 0.29 (250) |
| | $\sigma_g$ range | 0.04 - 0.34 | 0.08 - 0.74 | 0.06 - 0.26 | 0.08 - 0.38 | 0.16 - 0.51 |
| | Median | 0.14 | 0.40 | 0.27 | 0.21 | 0.26 |
| **Quantitative percent (geometric mean & standard deviation) of dry soil mass** | | | | | | |
| **TOC** | $\bar{X} \pm \sigma$ (n) | 0.58 ± 0.72 (268) | 0.91 ± 1.0 (1666) | 0.33 ± 0.30 (2379) | 0.40 ± 0.47 (1424) | 1.5 ± 0.9 (69) |
| | Median | 0.3 | 0.6 | 0.3 | 0.3 | 1.5 |

(1) This USDA semi-quantitative scale of textural clay proportions is defined by x-ray peak height (counts/s) according to: 1 = very small (< 110); 2 = small (110-360); 3 = medium (360-1120); 4 = large (1120-1800); 5 = very large (>1800). Montmorillonite is a 2:1 smectite clay. Column and row titles are emboldened to guide readers' eye.

goethite and gibbsite dominate in subsurface soil (Figs. 2B and 4C). Whereas the ferric solids, which are subject to reductive dissolution (SI III Extended Geochemical Summary), are most concentrated in the oxygen-rich vadose zone (Fig. 4D), the aluminum solids (not subject to reductive dissolution) present considerable electrostatic charge through the vadose zone and into the aquifer (Fig. 4E).

Figure 5 presents a heat map surveying relationships in distribution of PFAS with composite surface and vadose subsurface geochemical properties. In overarching perspective, Fig. 5 depicts two prominent clusters, (i) the long-chain PFAS are clustered on the air-water interface, total organic carbon and HFO concentration, whilst (ii) short-chain PFAS are strongly clustered on the aluminum-oxide mineral(oid) and clay concentrations, and electrostatic charges. We estimated air-water interfacial area using the methods of Brusseau[44] (Supplementary Eqs. 17 and 18) and the vertical position of soil samples above the free-standing water table (SI I Supplementary Methods). Consistent with the findings of Brusseau[32], our long-chain PFCAs and PFSAs ($CF_2 \geq 8$ for our data) cluster on air-water interface more prominently than for shorter chains (Fig. 5). In addition, our long-chain PFAS cluster on total organic carbon and total nitrogen (Fig. 5), which largely are composed of NOM that usually concentrates in surface soils as it does in our study sites (Supplementary Figs. 20–22). Long-chains also cluster with HFO concentration but not HFO charge, likely reflecting HFOs concentrating in surface soil (Fig. 4C) and the well documented intimate relations of incipient ferric oxides with NOM[45,46]. Inverse long-chain correlations on the clay fractions in Fig. 5 reflect the dominant presence of long-chains at the sandy surface, but authigenic crystalline oxides and clays accumulating in the subsurface. Whereas the long-chain PFAS clustering on air-water interface and organic matter comports with widely held conceptual models, the short-chain PFAS ($CF_2 \leq 7$) clustering heavily on concentration of oxide minerals and clays, and the electrostatic charges of HAO, gibbsite, and kaolinite, has not been widely reported as a dominant sink in subsoils (Fig. 5). HAO, gibbsite and kaolinite share positive Al surfaces at pH ≤ 7 (Fig. 4B and Supplementary Tables 10 and 11), thereby directly attracting the negative charge of most recalcitrant PFAS[13], and these authigenic mineral(oid)s have been documented to be nearly ubiquitous in US soil profiles (Table 1 and Supplementary Tables 19 and 20). Regarding short-chain clustering on illite, which bears permanent negatively charged surfaces, anionic PFAS sorption on negatively charged illite facets might be fostered by bridging of divalent solution cations[47,48] under the low-NOM conditions characteristic of subsurface soils, sorbing at the pH-dependent broken-bond sheet edges of illite[49], and/or diffusion into the clay interlayer. In any case, such a strongly expressed cluster between short-chain PFAS and these globally distributed authigenic mineral(oid) concentrations/charges is highly suggestive of their prominent role in the terrestrial fate of PFAS, as well as other anionic anthropogenic contaminants, in the subsurface environments of the United States and elsewhere globally.

While the role of authigenic aluminum oxide mineral(oid)s in the terrestrial fate of PFAS and other organic contaminants has not been deeply explored, the role of these solids in the distribution of inorganic anions and trace metals has been studied for more than a century[29,43]. Modeling of inorganic ion sorption on metal oxides includes both electrostatic and intrinsic components[29,43]. In Fig. 5, short-chain PFAS loading on aluminum solids commonly falls in the same general significance level for both concentration and electrostatic charge, suggesting a potential intrinsic contribution of PFAS sorption on these

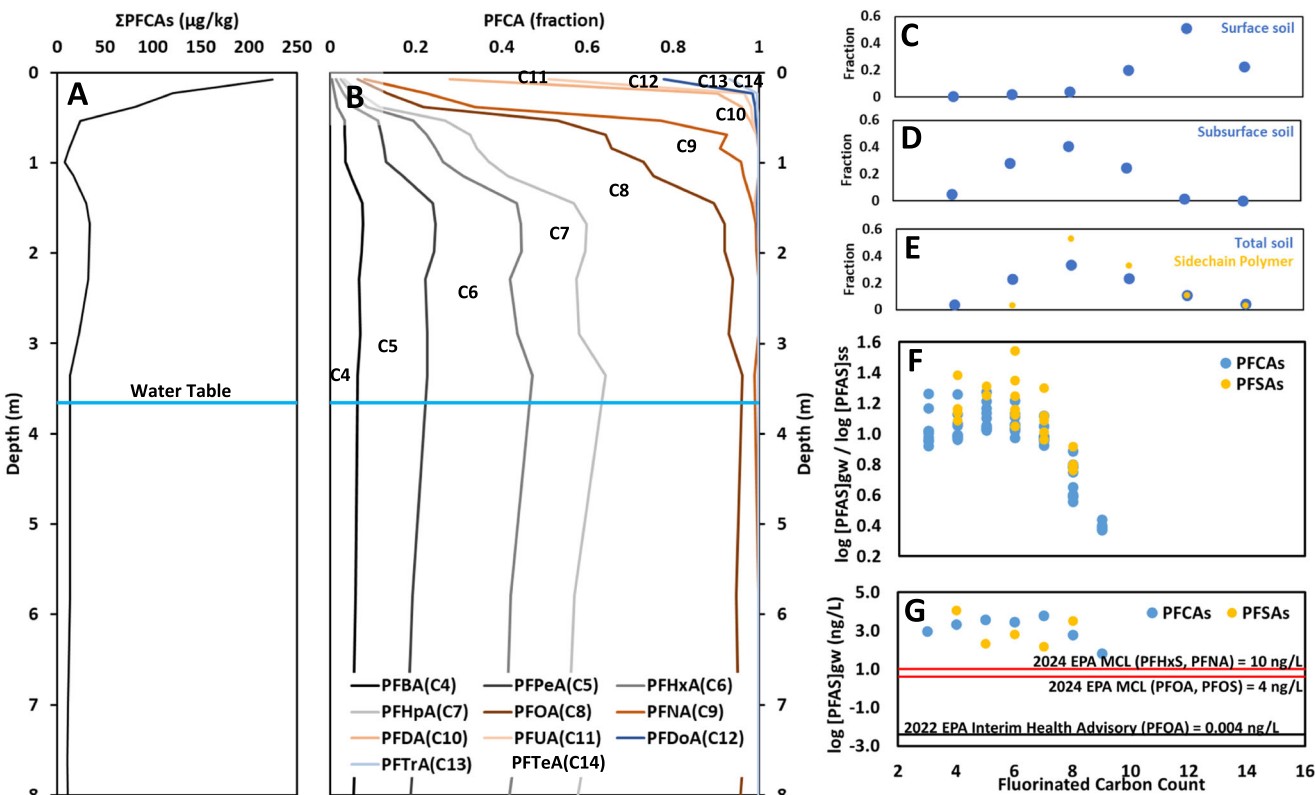

**Fig. 3 | Perfluorocarboxylates (PFCAs) in Field 1 soil as a function of depth.**
**A** ΣPFCAs remain highest at the soil surface more than 15 years after sludge application (**B**) The surface soils are dominated by chain-lengths C10–C14, but subsurface are dominated by shorter chain-lengths (C4–C8). When PFCAs are grouped according to their inferred intermediate precursor compounds, the FTOHs, surface soils have more long chains (**C**) and subsurface soils have more short chains (**D**) than typical sidechain fluorotelomer polymers, but summed over the full soil profile, the homologue distribution compares reasonably with the inferred ultimate precursor (**E**). **F** Because of relatively higher mobility of shorter-chain PFCAs and perfluorosulfonates (PFSAs), compared to the surface-soil source, groundwater is burdened with more short chain PFCAs than long, with an absence of detections for PFDA (C10) and longer compounds. **G** Even 1.5 decades after sludge application, groundwater still contains up to two orders-of-magnitude higher PFCAs than the 2024 EPA MCLs of 10 (PFHxS, PFNA) and 4 (PFOA, PFOS) ng/L.

solids. In early research on PFAS interactions with minerals, Hellsing et al. studied sorption of four PFAS to alumina (positive surface charge) and silica (negative surface charge) crystals and observed sorption to $Al_2O_3$ but not $SiO_2$, reflecting the role of electrostatic interactions we report here[50]. Studying sorption of PFOA and PFOS on amorphous (high surface area) $SiO_2$, Chang et al.[51] reported considerable non-linear second-order susceptibility, indicating intimate surface contact and suggesting a measurable intrinsic component for PFAS sorption on mineral surfaces. Bolstering this premise, Wahman et al.[52] studied the sorption selectivity, relative to chloride, of nine monovalent PFAS on three ion-exchange resins and uniformly found a 1- to 5- order-of-magnitude preference for PFAS sorption over $Cl^-$; if sorption were purely electrostatic, the large, diffusively charged PFAS would not be so strongly preferred over $Cl^-$. Unfortunately, unlike the chemically indifferent forces of the electrostatic interactions explored here, the intrinsic component of sorption is ion and surface specific, and therefore remains a challenge for future investigations.

Despite that ferric oxides have been shown to electrostatically sorb PFAS[53] and exhaustively documented to sorb inorganic anions[30,43], in our dataset characterizing full soil profiles, PFAS do not cluster upon ferric-oxide charge (Fig. 5), likely largely as a consequence of the restricted vertical distribution of ferric oxides dominantly at shallow B horizon depths (Figs. 2B and 4D, and Supplementary Fig. 5). In contrast, aluminum-oxide charge achieves its maximum at typical B-horizon depths but sustains considerable charge contributions through most of the aquifer column (Figs. 2B and 4E, and Supplementary Fig. 5) thereby offering governance of PFAS distribution throughout the sampling interval. This restricted vertical distribution

of ferric oxides at our study sites, relative to aluminum oxides, might be related to the potential for reductive dissolution of ferric oxides under low-oxygen conditions (Supplementary Fig. 6), for which aluminum oxides are not susceptible (SI III Extended Geochemical Summary). However, ferric oxides can be vertically distributed much more extensively in some settings[54]. Regardless, these results highlight the need to document the distribution of potential sorbents as well as their capacity in assessing fate of organic contaminants in the subsurface.

## Discussion

Soils constitute a considerable global-scale reservoir for PFAS[55], with biosolids and sludge land application among the top sources of PFAS introduction to the terrestrial environment[56]. At present, the terrestrial mobility of PFAS and other organic contaminants commonly is characterized with distribution constants ($K_d$ or $K_{oc}$ values) generated by sorption experiments with surface soils[57], offering insight into partitioning of PFAS within this thin mantle at land's surface. However, pedogenic processes generally render subsurface settings, starting as little as centimeters below the surface, drastically different than surface soils in NOM content and mineral assemblage (SI III, Supplementary Tables 19–21 and Supplementary Fig. 6), as observed in our study sites (Supplementary Tables 7 and 8, and Supplementary Figs. 20–22). So the applicability of these surface-soil values for accurately modeling fate of PFAS in the subsurface is dubious. Consistent with existing conceptual models, in our study spanning full soil columns, long-chain PFAS ($CF_2 \geq 8$) mobility appears to be controlled by a combination of air-water interface[32] and organic matter[31] in surficial soil. In contrast, short-chain PFAS ($CF_2 \leq 7$) mobility in the subsurface

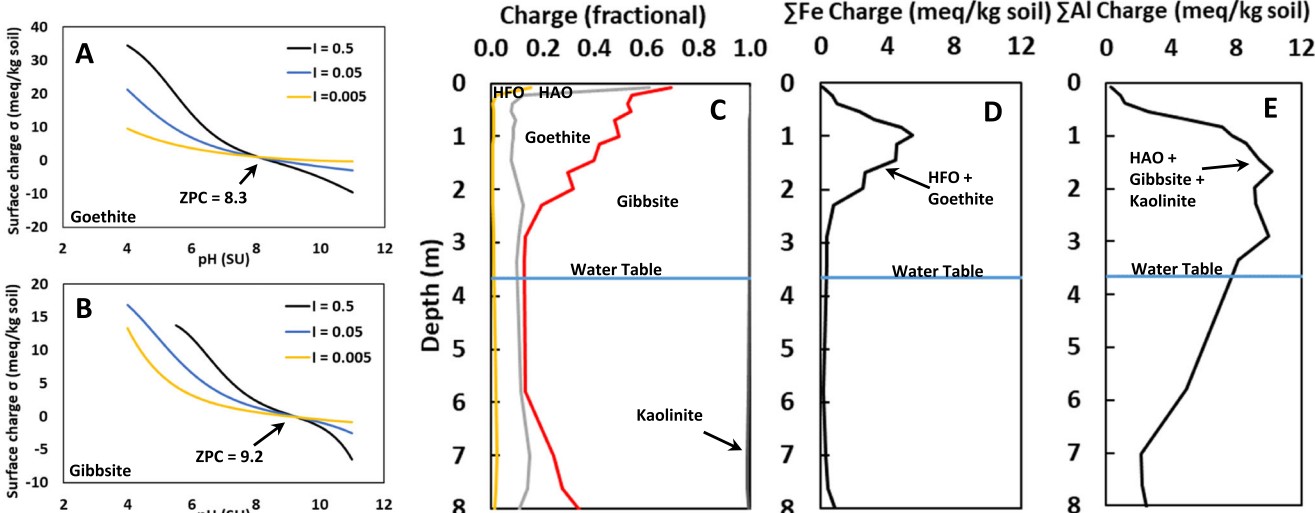

**Fig. 4 | pH- and ionic-strength –dependent electrostatic charge of authigenic (hydr)oxide Fe and Al mineral(oids).** The electrostatic surface charge of incipient, amorphous hydrous ferric oxide (HFO) and aluminum oxide (HAO), and the crystalline goethite (FeO(OH)) (**A**) and gibbsite ($Al(OH)_3$) (**B**) varies as a function of pH and ionic strength according to Supplementary Eq. 16, positively charged at low pH, diminishing to, and converging at, the zero-point of charge (ZPC). **C** Using measured concentrations of these mineral phases, and pH and estimated ionic strength (Supplementary Eqs. 17 and 18), the fraction of the total positive electrostatic surface charge is shown in our colluvial study site, showing that the incipient HFO and HAO are considerable in the surface soil, then the crystalline phases become more prominent in the subsurface, with goethite mostly in the vadose zone and gibbsite generally dominant near and below the water table (SI III Extended Geochemical Summary). **D, E** Summing these surface charges, the sandy surface soil bears relatively little mineral-derived positive electrostatic surface charge and increases to higher capacity in the soil horizons of illuviation, with considerable surface charge maintained through the shallow aquifer for the Al (hydroxides) +kaolinite (**E**).

| Field Site | Perfluoroalkyl Substance (#CF₂) | Air-Water Interface (Eq.S20) | Organics | | Ferric mineral(oid)s | | | | Aluminum mineral(oid)s | | | | Clay minerals | | | Textural fractions | | | |
|---|---|---|---|---|---|---|---|---|---|---|---|---|---|---|---|---|---|---|---|
| | | | TOC | Total N | Extn Hydrous Fe Oxide Conc. | Extn Hydrous Fe Oxide Charge | X-ray Goethite Conc. | X-ray Goethite Charge | Extn Hydrous Al Oxide Conc. | Extn Hydrous Al Oxide Charge | X-ray Gibbsite Conc. | X-ray Gibbsite Charge | X-ray Kaolinite Conc. | X-ray Kaolinite Charge | X-ray Illite Conc. | Textural Clay | Textural Silt | Textural Sand | X-ray Quartz Conc. |
| Field 1 & 2 | PFBA(3) | -0.21 | -0.25 | -0.21 | -0.19 | 0.06 | -0.24 | -0.23 | 0.46 | 0.39 | 0.23 | 0.25 | 0.22 | 0.25 | 0.35 | 0.32 | 0.23 | -0.45 | -0.19 |
| | PFPeA(4) | -0.31 | -0.34 | -0.27 | -0.25 | 0.09 | -0.18 | -0.16 | 0.44 | 0.45 | 0.31 | 0.34 | 0.32 | 0.35 | 0.46 | 0.40 | 0.14 | -0.50 | -0.28 |
| | PFHxA(5) | -0.38 | -0.44 | -0.31 | -0.36 | 0.12 | -0.23 | -0.16 | 0.43 | 0.58 | 0.33 | 0.45 | 0.36 | 0.47 | 0.52 | 0.45 | 0.13 | -0.55 | -0.31 |
| | PFHpA(6) | -0.36 | -0.44 | -0.30 | -0.36 | 0.11 | -0.25 | -0.19 | 0.47 | 0.58 | 0.31 | 0.41 | 0.34 | 0.43 | 0.46 | 0.44 | 0.14 | -0.55 | -0.28 |
| | PFOA(7) | 0.07 | -0.24 | -0.25 | -0.01 | 0.41 | -0.30 | -0.27 | 0.74 | 0.56 | 0.15 | 0.16 | 0.03 | 0.08 | -0.04 | 0.29 | 0.43 | -0.51 | 0.00 |
| | PFNA(8) | 0.36 | 0.20 | 0.09 | 0.20 | -0.09 | 0.06 | -0.06 | 0.33 | -0.20 | -0.14 | -0.40 | -0.20 | -0.41 | -0.30 | -0.05 | 0.18 | -0.02 | 0.17 |
| | PFDA(9) | 0.81 | 0.77 | 0.60 | 0.76 | -0.12 | -0.39 | -0.52 | -0.19 | -0.82 | -0.70 | -0.90 | -0.76 | -0.90 | -0.83 | -0.73 | 0.47 | 0.67 | 0.74 |
| | PFUA(10) | 0.89 | 0.88 | 0.76 | 0.80 | 0.28 | -0.43 | -0.66 | -0.51 | -0.66 | -0.77 | -0.74 | -0.84 | -0.62 | -0.83 | -0.88 | 0.41 | 0.83 | 0.81 |
| | PFDoA(11) | 0.84 | 0.90 | 0.82 | 0.77 | 0.15 | -0.53 | -0.55 | -0.63 | -0.70 | -0.81 | -0.75 | -0.83 | -0.65 | -0.80 | -0.91 | 0.38 | 0.89 | 0.83 |
| | PFOS(8) | 0.52 | 0.42 | 0.33 | 0.30 | -0.32 | 0.09 | -0.06 | -0.11 | -0.60 | -0.35 | -0.62 | -0.35 | -0.57 | -0.46 | -0.33 | 0.11 | 0.33 | 0.32 |
| Field 1 | PFBS(4) | -0.61 | -0.56 | -0.43 | -0.58 | 0.10 | -0.13 | -0.04 | 0.15 | 0.62 | 0.50 | 0.63 | 0.62 | 0.67 | 0.76 | 0.48 | 0.21 | -0.56 | -0.54 |
| | PFPeS(5) | -0.78 | -0.86 | -0.77 | -0.74 | 0.33 | 0.16 | 0.16 | 0.28 | 0.89 | 0.75 | 0.78 | 0.90 | 0.91 | 0.92 | 0.82 | -0.75 | -0.81 | -0.87 |
| | PFHxS(6) | -0.53 | -0.32 | -0.19 | -0.37 | 0.05 | -0.36 | -0.27 | -0.04 | 0.42 | 0.31 | 0.48 | 0.42 | 0.52 | 0.61 | 0.25 | 0.35 | -0.33 | -0.32 |
| | PFHpS(7) | -0.03 | -0.11 | 0.00 | -0.22 | -0.40 | -0.05 | -0.15 | 0.43 | 0.10 | 0.12 | -0.01 | 0.24 | 0.08 | 0.23 | 0.25 | 0.24 | -0.32 | -0.20 |
| | PFOS(8) | 0.69 | 0.44 | 0.40 | 0.48 | -0.44 | 0.25 | -0.04 | 0.43 | -0.66 | -0.33 | -0.78 | -0.38 | -0.77 | -0.65 | -0.18 | 0.09 | 0.18 | 0.32 |

**Fig. 5 | Heat map of nominal correlation coefficients for PFAS concentrations (rows) with geochemical properties (columns).** Geochemical properties are grouped according to close functional relationships. Perfluorocarboxylates (PFCAs) are ordered by increasing $CF_2$ chain-length, for combined Field 1 and 2 data, and perfluorosulfonates (PFSA) for Field 1 (most PFSAs were undetected in Field 2), above the water table. Increasing intensity of red coloration denotes progressively more significant positive nominal correlations (based on statistical $p$-values; yellow $p \leq 0.10$, orange $p \leq 0.05$, dark orange $p \leq 0.01$), and blue intensity (light to dark from $p \leq 0.10$ to $p \leq 0.01$) similarly reflects inverse negative nominal correlations as a consequence of inverse covariance with positively correlated properties in natural soil columns.

terrestrial setting is controlled by electrostatic, and perhaps intrinsic, sorption on a limited number of authigenic mineral(oid) surfaces. This affinity that we describe here of short-chain PFAS with a few authigenic minerals, which are nearly ubiquitously distributed in the subsurface (Supplementary Tables 19–21), offers implications both local and large-scale, immediate and forward thinking.

At the local and immediate scale, these results suggest that studies assessing the fate of PFAS and other organic contaminants in the subsurface might be better designed by including $K_d$ values determined with subsurface soils in addition to surface soils, such as B-horizon soil samples for deeper-rooted agricultural vegetation and C horizons for migration to and within aquifers. Moreover, given the sensitivity of the anion-sorption complex to pH and ionic strength (Fig. 4), ideally these sorption experiments ought to be conducted under measured subsurface pH and ionic-strength conditions. It is noteworthy that the short-chain PFAS correlation with charge generally exceeds that with concentration (Fig. 5) suggesting that non-specific electrostatics constitute a large fraction of total sorption capacity, perhaps tempering the need to determine mineral assemblages in quest of greater precision in many efforts.

In terms of large-scale patterns, our present study suggests that the recent transition to use of relatively mobile short-chain PFAS[13] may result in more dynamic future cycling of PFAS in the environment than for the historically used long-chain PFAS, consistent with the finding of our recent independent study[6]. Spatially, northern US (e.g., recent glacially derived soils) and less-weathered soils/settings tend to have lesser concentrations of pH-dependent (hydr)oxide minerals than more weathered (Table 1, Supplementary Tables 19–21 and Supplementary Fig. 6). Offsetting these lower mineral concentrations, lesser weathered soils can have higher ionic strength (Fig. 2A) which fosters higher surface charge per mass of these phases (Fig. 4A, B) and less weathered soils tend to have more 2:1 minerals (Fig. 2A, Table 1 and Supplementary Fig. 6) which also concentrate PFAS (Fig. 5). In the vertical dimension, aquifer settings, wherein dissolved oxygen commonly is limited, might be more dependably rich in aluminum (hydr) oxides as potential sorbents than ferric (hydr)oxides which are subject to reductive dissolution (SI III, Supplementary Figs. 6 and 7).

Regarding thinking forward, the precision of subsurface partitioning might be further enhanced in future studies with specific-sorption experiments (i.e., $K^{int}$) of individual PFAS on the limited array of authigenic minerals discussed here and studies of PFAS $K_d$ values in A-,B-, and C-horizon soils of selected soil orders. Still more broadly in terms of contaminants, the fundamental nature of these partitioning drivers, i.e., innate electrostatics and/or hydrophobicity, suggest a similar mode of subsurface mobility for other anthropogenic compounds that commonly make their way to soils, such as pesticides and pharmaceuticals[22]. The efficacy of these reversible sorption processes to serve as a terminal subsurface repository will depend upon the environmental transformation rate for these contaminants, with persistent contaminants like PFAS continuing to cycle through the environment in the long run.

## Methods

All study methods are summarized here and described in detail in the on-line Supplementary Information (SI).

### Sampling

Two agricultural fields in the South Carolina coastal plain consisting of marine sediments, that had received sludge from the now defunct Galey and Lord textile mill, were selected to study the fate of PFAS contaminants present in the sludge. At each field, a Geoprobe drill rig was used to drive cores through the soil profile and underlying water-table aquifer to a clay aquiclude. In addition to collecting solid core samples, at each site, two temporary water wells were installed with screens located near the shallow water table (at 3.7 to 4.6 m) and the deep aquiclude (-18 m), followed by water sampling for analysis.

### Analysis

All sampled soil and aquifer solids (-50 at each field site) were analyzed for conventional soil-characterization parameters (e.g., pH, cation exchange capacity, base saturation, lime requirement, Mehlich metals, water-extractable anions, total organic carbon, total nitrogen, and particle size) at the University of Georgia cooperative extension laboratories. Soil and aquifer solids were subjected to bulk x-ray diffraction analyses with semi-quantitation of major mineral phases at the University of Georgia Department of Geology. Solids samples were subjected to selective extractions to quantitate the amorphous hydrous-oxide mineraloid fractions of ferric iron, aluminum, and manganese. Solids samples were analyzed for PFAS according to the ASTM D7968 method and water samples were analyzed using the ASTM 7979 method, all using liquid-chromatography, tandem mass spectrometry. Nontargeted analysis was performed on select extracts by liquid-chromatography high resolution quadrupole time-of-flight mass spectrometry.

### Summarizing content of minerals ubiquitously present in US soils

Mineralogy data of US soils were obtained via on-line query of the USDA Natural Resources Conservation Service (NRCS) National Cooperative Soil Survey (NCSS) Soil Characterization Database. Data on major soil authigenic minerals, quartz, amorphous solids of ferric iron and aluminum, and total organic carbon were extracted and grouped by USDA Soil Order and major pedogenic horizons of A, B, and C. The resulting summary of US soils consisted of >60000 data points.

### Calculation of variable electrostatic charge

For the common soil mineral(oid)s that possess a variable electrostatic charge, including goethite, gibbsite, kaolinite, HAO and HFO, we calculated charge for all solids samples as a function of pH and ionic strength using a generalized two-layer model[29,43]. Necessary inputs for these calculations were culled from literature and are summarized as Supplementary Table 5. This approach yielded -500 values of electrostatic charge (-100 samples × 5 mineral(oid)s) (SI).

### Calculation of vadose moisture content and air-water interface

We calculated air-water interfacial area of the soil-column samples using the median-particle approach of Brusseau[44]. To populate inputs for this equation, we used our measured USDA soil textures. We used literature values of median particle diameter reported for each USDA soil texture[58]. We assumed equilibrium water tension in the vadose zone relative to elevation above the water table, and employed the Boltzmann Distribution equation, parameterized for each USDA soil texture[59], allowing us to estimate air-water interfacial area for all vadose samples.

### Data availability

The processed data are available at the EPA Environmental Dataset Gateway (https://catalog.data.gov/dataset/), under https://doi.org/10.23719/1532057[60]. The data generated in this study are provided in the Supplementary Information.

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

## Acknowledgements

The authors thank Jason Nemecek and Kyle Stephens of the United States Department of Agriculture (USDA) Natural Resources Conservation Service (NRCS) for their assistance with the National Soil Database, David Dzombak for supplying electrostatic data from MINEQL, Brian Schumacher and Eric Weber of EPA/ORD for helpful reviews, Mark Brusseau of the University of Arizona for discussions on air-water interface calculations, and the South Carolina private landowners for allowing access to conduct multiday sampling. Du Yung Kim is supported by an Oak Ridge Institute for Science and Education (ORISE) research fellowship award at the Center for Environmental Measurement and Modeling at EPA/ORD in Athens, Georgia. Mention of trade names or commercial products does not constitute endorsement or recommendation for use. The views expressed in this article are those of the authors and do not necessarily represent the views or policies of the U.S. EPA.

## Author contributions

J.F. and J.W.W. proposed the research and developed the study design. M.G.E., J.F., J.C.W., M.G., M.P.N., K.S., B.C.S., and J.W.W. conducted field work and collected environmental samples. M.G.E., B.A., D.B., D.A.G., W.M.H., D.Y.K., and J.W.W. performed the PFAS analyses, O.A., P.A.S., and S.B.C. performed the geochemical analyses, J.W.W. performed geochemical modeling, and M.G.E. and M.C. performed the statistical analyses. M.G.E. and J.W.W. drafted the manuscript. All authors edited the manuscript and contributed to the results.

## Competing interests

The authors declare no competing interests.
