## [Transparent Peer Review file · Nature Communications]

Mineralogical controls on PFAS and anthropogenic anions in subsurface soils and aquifers

Corresponding Author: Dr John Washington

Version 0:

Reviewer comments:

Reviewer #1

(Remarks to the Author)

The manuscript titled "Controls on the Fate of PFAS and Anthropogenic Anions in the Terrestrial Subsurface" presents a detailed examination of the mechanisms governing the mobility and fate of PFAS in soil and aquifer environments. It highlights the significant role of mineralogy, particularly authigenic minerals, in adsorbing PFAS, thereby influencing their environmental behavior and distribution.

Strengths:

The study's use of a wide array of geochemical and mineralogical analyses across different soil layers enhances the depth and reliability of the findings. This robust methodological framework provides a solid basis for the conclusions drawn. The application of advanced geochemical modeling to predict PFAS interactions with soil minerals adds a valuable dimension to the research, offering insights that are both predictive and explanatory. The detailed discussion on the impact of specific minerals like kaolinite and gibbsite on PFAS sorption is a strong point, providing clarity on the molecular interactions that govern PFAS mobility in subsurface environments. The manuscript effectively uses data to challenge and refine existing theories on PFAS behavior, particularly concerning their interactions with different soil minerals.

Weaknesses:

While the data is generally well presented, some figures and tables are densely packed with information. Simplifying these or providing supplementary online material could help maintain the reader's focus on key results. The manuscript could enhance its scope by incorporating more discussion on biological factors affecting PFAS fate, such as microbial degradation of precursors, which are equally crucial in subsurface environments. While the study presents detailed data, the generalization of results across varied environmental conditions could be better addressed to clarify the scope of application of the findings. The heavy use of technical jargon and complex descriptions may make the paper less accessible to a broader audience for Nature Communications. Please include a nomenclature section to aid understanding. The manuscript's references are up-to-date; however, incorporating discussions on 'older' but relevant studies, such as Xiao et al., 2015, on PFAS in surface and subsurface soils could enhance its thoroughness. Although the study provides a comprehensive current assessment, it could offer more explicit suggestions for future research directions, particularly in addressing the gaps identified in PFAS behavior under varying environmental conditions.

Reviewer #2

(Remarks to the Author)

General comments:

1. This dataset is significant and of high quality. I also think that its inclusion of impacts from the textiles industry (not a major focus of the paper) is quite unique, and I wonder if there are aspects of the release type (textiles, side chains, etc.) that have yet to be explored in the manuscript. I think there's a lot that could be done with these data, and I'm excited to see where the authors take the story based on reviewer feedback.

2. The significance of this study relative to prior work really needs to be clarified. As currently written, it is difficult to ascertain what this study contributes over prior work. Is it a conceptual model of PFAS sorption in the vadose zone and saturated zones with depth? Is it confirmation of a conceptual model that already existed with the extensive dataset the authors are presenting? There are a number of prior studies focused on PFAS sorption to minerals. The authors cite some of it (mostly in the supplementary material), but in my view the outcomes of this prior work were not thoroughly considered and were not used to identify a clear knowledge gap that is addressed here. It would be helpful if the authors outlined a clear conceptual model of the role of minerals in PFAS sorption based on outcomes of prior work, identified clearly knowledge gaps, and then provided clear hypotheses or objectives for the current study.

In case they are helpful, here are a few additional papers related to PFAS interactions with minerals that I'm not sure were considered:

- <https://doi.org/10.1016/j.chemosphere.2016.06.016>
- <https://doi.org/10.1021/acs.est.0c01646>
- <https://doi.org/10.1021/acs.est.2c01054>

3. Another aspect of significance that needs to be clarified is the importance of PFAS interactions with minerals in governing the overall distribution of PFAS. Showing correlations of PFAS with various mineral components is useful and interesting, but it does not inform the magnitude of the interactions of PFAS with those mineral phases (e.g. the magnitude of equilibrium sorption coefficients [K_d] values). Studies have shown that equilibrium sorption coefficients of PFAS to mineral phases can be quite low, and overall, the interactions of short-chain PFAS with any phase are quite low. So will these mineral phases really lead to a meaningful change in PFAS transport via sorption? Could we achieve similar predictions of transport, if we just assumed that there was no sorption to these phases? What about the role of other processes like matrix diffusion into/out of clays? I suspect the latter would be much more important in influencing PFAS transport over sorption to the minerals.

4. This study is very geology-oriented, uses a lot of the associated terminology, and it seems written for a geology audience over the type of broad readership often targeted by a journal such as Nature Communications. It will be difficult for a broader scientific audience to understand, and currently the relevance of this topic to a broad scientific audience isn't clear. PFAS are contaminants that are of global concern, but as written this manuscript seems better suited to a journal more focused on hydrogeology or environmental engineering. Clarifying the significance (as noted above) might go a long way to address this, but I also think that the final paragraphs of the paper need to do more to address the significance to a multidisciplinary readership.

Below are just a few, additional line by line comments.

Lines 17-31 (the abstract) are all background and a brief summary of the study objective. I suggest including a description of the significant outcomes of the study. Due to word limitations this may necessitate reducing the amount of background.

Line 42: Minor comment that "and the like" is a vague way to end the sentence. Plus this information is unsupported by references.

Line 50: Why only advection? Why not dispersion, back-diffusion, and other mechanisms?

Line 64 mentions fluorotelomer polymers, and I think it would be beneficial to denote that these are also PFAS (i.e., polymeric PFAS) that can degrade to non-polymeric PFAS. As written it implies that these are polymers (not PFAS) that can degrade into PFAS.

Lines 78-80: Please consider revising as this is not a complete sentence and it is unclear what the authors are trying to say.

Lines 117-135: Some of this information was already mentioned in the introduction, and this is just more general background on PFAS. If this space is to be used to focus on PFAS background, then I suggest using to define why this study is needed. So far the authors have simply said that current work focuses on surface soils, which are higher in Natural Organic Matter (NOM) than deeper soils. However, they haven't said anything about why understanding interactions with minerals in more weathered depths is important or what kinds of interactions are not already understood via prior work.

Line 240 discusses the strong correlation between negatively charged illite and concentrations of short-chain (and anionic) PFAS. Multiple hypotheses are given regarding the cause, but none address non-sorptive mechanisms such as the diffusion of PFAS into clay, which would presumably occur faster for short-chain PFAS due to low molecular weight.

Figure 1: Panel C could be better utilized. For example, A and B show details of composition that support the caption, but there is nothing in panel C that really represents what is described in the caption for that panel. I.e., the panel doesn't really reflect anything about, "Shallow-aquifer flow zones tend to weather less extensively than the vadose zone; in reducing waters, Fe (hydr)oxides may undergo reductive dissolution thereby not contributing to the potential exchange complex, but Al (hydr)oxides and kaolinite can sorb short-chain PFAS regardless of oxidation state."

Figure 3. There is an indication of a kaolinite line in Figure 3, panel C, but the kaolinite line is not visible.

Reviewer #3

(Remarks to the Author)

The authors present an interesting dataset combining field data from a large number of soil cores (~50/site) with extensive geological and geochemical information complemented with detailed PFAS analysis. These data and explaining the interesting trends in CF₂ chain length dependent distribution, especially in deeper, carbon depleted soil and aquifer layers, form the core of the manuscript, which is very interesting and valuable on its own.

However, the authors could have substantially increased the value of their work, had they further validated their interpretations with laboratory scale experiments (e.g. PFCA formation rates from precursors as well as sorption and desorption batch or column experiments at relevant conditions as partially suggested by the authors in line 296ff). This would have helped complementing their field data with some sort of kinetic modeling of PFCA fate (formation, transport sorption in dependence of water saturation, soil/aquifer composition, AEC, CEC, pH, ionic strength etc), to better contextualize, translate and enable extrapolation of the interesting findings shown in Figure 4. Such complementary activities would have given the study more weight and could have facilitated the translation of these findings to other sites and scenarios.

As the manuscript stands now, it is hard to say if the trends and findings on chain length dependent behavior and the importance of electrostatic interactions in carbon depleted matrices are a "just" really interesting observation from two field sites, or if -and how- these findings can be generalized and used for future modeling, risk assessment, and potentially prioritization of mitigation measures across the large number of potentially relevant sites mentioned by the authors in line.

Minor remarks:

Line 52-56: a bit of an overstatement on the lack of studies on minerals and NOM, as these aspects have been the focus of a substantial number of studies over the last decade(s), including for charged and ionizable compounds.

Line 62: a sentence contextualizing the focus on sidechain fluorotelomer within the larger family of PFAS may be useful for some readers.

Line 73: rephrase "in soils in our soils and soils"

Line 219: this paragraph could be extended somewhat.

Line 261: some systematic experiments would have been possible and highly appreciated to get a better understating on the sorption processes. Please also see my general comments.

Version 1:

Reviewer comments:

Reviewer #1

(Remarks to the Author)

The manuscript has undergone significant improvements, with additional information included and a thorough response to the prior comments, which have enhanced the clarity and depth of the study.

Overall, the authors have effectively addressed concerns about the dense packing of figures and tables. They have simplified the presentation and have relocated extensive data to supplementary material, which helps maintain focus on the key results.

By incorporating more discussion on the biological factors affecting PFAS fate, such as microbial degradation, the manuscript offers a more comprehensive view of the mechanisms governing PFAS behavior in subsurface environments.

The authors have made considerable efforts to simplify the language and structure of the manuscript, making it more accessible to a broader audience without sacrificing technical depth.

Given the comprehensive nature of the revisions and the manuscript's alignment with the journal's scope, particularly in terms of its interdisciplinary approach and potential for broad interest, I recommend this manuscript for publication.

Reviewer #2

(Remarks to the Author)

I would like to thank the authors for simplifying the use of technical jargon, improvements to figures, and improved framing of the contribution of their work. As I understand it, important contributions of this work include use of calculated, pH-dependent charge when correlating subsurface PFAS concentrations with geochemical parameters, an improved conceptual model of PFAS interactions with subsurface soils that includes the relative importance of various minerals, and an evaluation of the global ubiquity of these minerals based on the United States Department of Agriculture (USDA) soils database. I agree that these results offer important considerations for those evaluating PFAS fate and transport either through laboratory experiments or modeling. However, I think it is the application of these results that will be of most interest to a broad, multidisciplinary audience consistent with Nature Communications. For example, it would be great to see a quantitative analysis (e.g., fate and transport model- could be a simplified one) that demonstrates how consideration of these mineral-specific interactions influence the distribution of short-chain PFAS vs. a model that considers homogenous surface soil composition vs. (perhaps) a model that considers little to no sorption to minerals. Without that application, it is difficult to frame the importance of the authors' advancements to the PFAS sorption conceptual model. I have provided some point by

point feedback within my original comments and the authors' responses in the document below. There are some additional, minor comments unique to this version beneath this point by point response. All line numbers reference the track changes version of the manuscript.

Initial comments/author responses:

1. This dataset is significant and of high quality. I also think that it's inclusion of impacts from the textiles industry (not a major focus of the paper) is quite unique, and I wonder if there are aspects of the release type (textiles, side chains, etc.) that have yet to be explored in the manuscript. I think there's a lot that could be done with these data, and I'm excited to see where the authors take the story based on reviewer feedback.

This observation regarding textile industry is quite important and we plan to explore it more directly in future papers. While there are several other textile-PFAS papers, we are not aware of any with the amount of data we have reported, nor the focus on textile sludge, which we believe other mills have spread on land as well. We are continuing research on wastes from this textile mill at present.

Reviewer response: Thanks

2. The significance of this study relative to prior work really needs to be clarified. As currently written, it is difficult to ascertain what this study contributes over prior work. Is it a conceptual model of PFAS sorption in the vadose zone and saturated zones with depth? Is it confirmation of a conceptual model that already existed with the extensive dataset the authors are presenting? There are a number of prior studies focused on PFAS sorption to minerals. The authors cite some of it (mostly in the supplementary material), but in my view the outcomes of this prior work were not thoroughly considered and were not used to identify a clear knowledge gap that is addressed here. It would be helpful if the authors outlined a clear conceptual model of the role of minerals in PFAS sorption based on outcomes of prior work, identified clearly knowledge gaps, and then provided clear hypotheses or objectives for the current study.

Thank you for this important point. Having worked intensely on this for so long, we did not step back far enough to offer perspective. We have edited the abstract and added a paragraph to the introductory section, lines 65-73 of the track-changes manuscript in simple mark-up mode, to position our study in context of current practice. In this paragraph, we reference a highly cited Feature Article in ES&T, identifying pH-dependent charge as important, but not reporting a quantitative method to calculate this property, a limitation that our submission addresses. In addition, for our tabulated summaries of the USDA soil database (Table 1, Supplementary Tables 19-21), we have added total organic carbon (TOC) to illustrate how TOC is greatly diminished in subsurface horizons compared to the surface, thereby limiting its role in PFAS fate in the subsurface relative to the organic-rich surface soils.

Reviewer response: Lines 65-73 of the track changes manuscript do not appear to reflect any tracked changes, and do not appear to discuss the topics mentioned in the authors' response (e.g., pH-dependent charge).

In case they are helpful, here are a few additional papers related to PFAS interactions with minerals that I'm not sure were considered:

- <https://doi.org/10.1016/j.chemosphere.2016.06.016>
- <https://doi.org/10.1021/acs.est.0c01646>
- <https://doi.org/10.1021/acs.est.2c01054>

We thank the Reviewer for these great reference suggestions.

Reference <https://doi.org/10.1021/acs.est.0c01646> was previously considered and is cited in the manuscript. The other references have been added to the manuscript at Line 245-247 (simple markup): In early research on PFAS interactions with minerals, Hellsing et al. studied mineral sorption of four PFAS to alumina (positive surface charge) and silica (negative surface charge) crystals and observed sorption to Al₂O₃ but not SiO₂, suggesting a critical role of electrostatic interactions 50. And have been added to the Supplementary Information at Lines 1019-1025: Interestingly, simulations found significant variation in molecular clustering and coordination between the PFAS and the hydroxyl surfaces of kaolinite among molecules of different chain-length and functional group, suggesting short-chain sulfonate PFAS molecules experience more restrictive transport in environmental soils with a higher kaolinite compared to carboxylates 97. These studies could provide some mechanistic insight for the significantly higher correlations for short-chain sulfonates with kaolinite in Field 1 unsaturated media relative to short-chain carboxylates (Supplementary Fig. 23).

Reviewer response: Thanks

3. Another aspect of significance that needs to be clarified is the importance of PFAS interactions with minerals in governing the overall distribution of PFAS. Showing correlations of PFAS with various mineral components is useful and interesting, but it does not inform the magnitude of the interactions of PFAS with those mineral phases (e.g. the magnitude of equilibrium sorption coefficients [K_d] values). Studies have shown that equilibrium sorption coefficients of PFAS to mineral phases can be quite low, and overall, the interactions of short-chain PFAS with any phase are quite low. So will these mineral phases really lead to a meaningful change in PFAS transport via sorption? Could we achieve similar predictions of transport, if we just assumed that there was no sorption to these phases? What about the role of other processes like matrix diffusion into/out of clays? I suspect the latter would be much more important in influencing PFAS transport over sorption to the minerals.

Thank you for these thoughts. The K_d values for shorter PFAS with mineral phases indeed can be modest.

Regarding the mineral phases, a central point we make in our manuscript, is that numerous common soil minerals have pH- and ionic-strength dependent surface charges (see our new Figure 4, formerly Figure 3). To represent potential sorption capacity for these minerals under real-world conditions, sorption experiments ideally should be carried out at pHs and ionic strengths similar to that in the subsurface, commonly with pH < 5 and ionic strength (I) on the order of 0.05. Commonly laboratory experiments are carried out at higher pH values (closer to the minerals' zero point of charge) and lesser ionic strengths, potentially leading to understated values of K_d.

Regarding the PFAS compounds, as the reviewer knows, sorption varies as a sensitive function of chain length. This is evident in our Figure 3 (Figure 2 in our original submission), Panel A, which depicts a sharp drop-off in Σ PFAS to a minimum at ~1 m, near the base of eluviated horizons, then an increase to the subsurface maximum at 1.5 to 2 m, at depths typical of illuviated accumulation of authigenic minerals. This pattern also is reflected in Figure 3, Panel B, in that C5 through C7 also clearly exhibit local maxima at this depth interval. These details would not be predicted by a conceptual model of no sorption. In contrast, C4 shows no such maximum and, for these data, might be consistent with your question of whether assuming no sorption would be an effective conceptual model. Detailed knowledge of these behaviors are not well developed in natural systems and contributions like we report here are necessary for developing this knowledge.

On a larger-scale, even minor partitioning to mineral phases in the subsurface is important to understand because minor sorption, perhaps like C4, expressed over long flow-paths can lead to considerable accumulation. Such minor accumulation over longer flow-paths is important to appreciate in efforts to understand, for example: i) migration times of plumes, ii) magnitude of clean-up efforts, e.g., how long to expect pump-and-treat, and iii) potentially how to foster faster remediation, e.g., increasing pH to the ZPC, thereby diminishing sorption to the stationary phases.

Regarding diffusion into clays, we have added this as a possible mechanism in lines 231-235. While we definitely agree this is a potential mechanism of retention, whether this is a primary mode of sorption seems questionable: i) the PFAS we report upon are anions and the mineral surfaces are positively charged, and electrostatic attraction is a well-established fundamental phenomenon; ii) the perfluorinated chains of PFAS are mutually repulsive to and repulsed by higher dipole moments, and the hydrated diffuse interlayer of expanding smectite clays is well-ordered by the permanent electrostatic surface charge of smectite clays compared to the bulk aqueous phase, so the hydrophobic PFAS chains might be more repulsed by the electrostatically ordered diffuse layer adjacent to clay surfaces than in the bulk water; iii) none of the minerals we report upon here are expanding smectite clays, so interlayer spacing is quite restricted; iv) correlations in Figure 5 (previously Figure 4) for the large majority of PFAS-mineral pairs are more significant with electrostatic charge than with mineral concentration; and v) for the shorter chains, PFBA through PFOA, correlations mostly increase with increasing chain-length, and diffusion into clay interlayers, to the extent it is important, would seem likely to favor shorter chains due to steric hindrance of longer chains in tight interlayers and repulsion of longer chains in the highly electrostatically ordered interlayer.

Reviewer response: The authors indicate that minor sorption can lead to considerable accumulation, but this is purely conceptual, and I think that constitutes a remaining weakness of the manuscript. The authors have not done any quantitative analysis to demonstrate the degree to which these mineral interactions could influence fate and transport. It seems like it would be feasible to use site conditions and simplifying assumptions to model how transport looks with and without these charge-based interactions. It doesn't necessarily need to be a model that is fitted to field data, just application of the study outcomes in a more quantitative way that would demonstrate, for example, migration of PFBA over a 20 year period with and without these mineral interactions.

4. This study is very geology-oriented, uses a lot of the associated terminology, and it seems written for a geology audience over the type of broad readership often targeted by a journal such as Nature Communications. It will be difficult for a broader scientific audience to understand, and currently the relevance of this topic to a broad scientific audience isn't clear. PFAS are contaminants that are of global concern, but as written this manuscript seems better suited to a journal more focused on hydrogeology or environmental engineering. Clarifying the significance (as noted above) might go a long way to address this, but I also think that the final paragraphs of the paper need to do more to address the significance to a multidisciplinary readership.

Thank you, this is a good point. We have made substantial changes throughout the manuscript, taking care to make the article more readable for a general audience. Further discussion and more technical descriptions are reserved for the Supplementary Information.

We also agree that PFAS are of global concern. And much of the global fate of PFAS is governed by processes in the subsurface. We hope with our modifications to the manuscript, we can bring to light processes important to the fate of PFAS globally, without overburdening the reader with geochemical jargon.

Reviewer response: Thanks

Below are just a few, additional line by line comments.

Lines 17-31 (the abstract) are all background and a brief summary of the study objective. I suggest including a description of the significant outcomes of the study. Due to word limitations this may necessitate reducing the amount of background. We thank the Reviewer for this suggestion. We have reworded the abstract substantially to reduce background and include more detail on the study outcome.

Lines 19-30 now read:

Per- and polyfluoroalkyl substances (PFAS), like many anthropogenic compounds, migrate into the environment through

various means, e.g., soil-amendment impurities and ambient atmospheric deposition, potentially resulting in vegetative uptake and migration to groundwater. Existing approaches for modeling sorption of PFAS and other compounds commonly include treating soil as an undifferentiated homogeneous medium, with distribution constants (e.g., K_d , K_{oc}) generated for individual compounds empirically using surface soils. Taking into consideration the limited mineral variety expected in weathered geologic media, mobility of PFAS and other anthropogenic compounds can be better understood by accounting for these predictable mineral assemblages. Here we explore the role of minerals and electrostatic sorption in controlling PFAS mobility in subsurface soils and shallow aquifers at agricultural sites following heavy contamination by measuring geochemical parameters and PFAS, and calculating pH-dependent mineral surface charges through full soil and aquifer columns. Furthermore, we report on the ubiquitous distribution of these minerals in U.S. soils.

Reviewer response: Thank you for the edits to your abstract; however, I still do not see any mention of the primary study outcomes within the edited version. For example, "Here we report the role of minerals and electrostatic sorption in controlling PFAS mobility in the subsurface soils and shallow aquifers at agricultural sites following heavy contamination by measuring geochemical parameters and PFAS, calculating pH-dependent mineral surface charges through full soil and aquifer columns," is a summary of approach. Not an outcome.

Line 42: Minor comment that "and the like" is a vague way to end the sentence. Plus this information is unsupported by references.

Thank you for the comment; we removed "and the like" from the sentence.

Lines 40-41 now reads: Often these biosolids bear a witches' brew of our societal waste, e.g., pharmaceuticals, PFAS, and plasticizers 2.

Reviewer response: Thanks

Line 50: Why only advection? Why not dispersion, back-diffusion, and other mechanisms?

Thank you for pointing that out; we have made the statement more general to include other groundwater solute transport mechanisms.

Lines 44-48 now reads: Released to the land's surface, these pesticides, and manure-, biosolids- and atmospheric-borne chemicals interact intimately with soils, which initially act as sinks from the aqueous phase, serving as sizable reservoirs, and ultimately time-delayed sources for moderated concentrations of recalcitrant contaminants, perhaps to be accumulated in vegetation or soil organisms 6 or to leach through the vadose zone and migrating to groundwater 7,8.

Reviewer response: Thanks

Line 64 mentions fluorotelomer polymers, and I think it would be beneficial to denote that these are also PFAS (i.e., polymeric PFAS) that can degrade to non-polymeric PFAS. As written it implies that these are polymers (not PFAS) that can degrade into PFAS.

We have added the additional information to make this clearer to the reader.

Lines 56-59 now reads: Among PFAS uses, the widespread industries of textile¹⁵, paper¹⁶, and carpet¹⁷ milling operations historically have used polymeric PFAS, e.g., sidechain fluorotelomer polymers, to treat their products for anti-staining properties 18.

Reviewer response: Thanks

Lines 78-80: Please consider revising as this is not a complete sentence and it is unclear what the authors are trying to say. We have modified this sentence for clarity and grammar.

Lines 80-83 now reads: Here we report PFAS distribution patterns through the subsurface, document the ubiquitous distribution of authigenic minerals in soils across the United States, and explore how these globally distributed minerals and other geologic soil parameters control distribution and fate of PFAS in the subsurface (SI II).

Reviewer response: Thanks

Lines 117-135: Some of this information was already mentioned in the introduction, and this is just more general background on PFAS. If this space is to be used to focus on PFAS background, then I suggest using to define why this study is needed. So far the authors have simply said that current work focuses on surface soils, which are higher in Natural Organic Matter (NOM) than deeper soils. However, they haven't said anything about why understanding interactions with minerals in more weathered depths is important or what kinds of interactions are not already understood via prior work.

We thank the reviewers for drawing our attention to this section. We removed this entire section from the manuscript; information which has already been mentioned was removed and new details were moved to the introduction and following section. We have added a paragraph to the introductory section, lines 65-73, to make clear that the subsurface is composed of minerals, that contaminant-mineral interaction is conceptually understood to be important in subsurface fate, but that specific minerals in real-world soil bodies, and quantitative expression of mineral properties that might affect fate, are not as well defined.

Reviewer response: Thanks

Line 240 discusses the strong correlation between negatively charged illite and concentrations of short-chain (and anionic) PFAS. Multiple hypotheses are given regarding the cause, but none address non-sorptive mechanisms such as the diffusion of PFAS into clay, which would presumably occur faster for short-chain PFAS due to low molecular weight.

We agree that this potential mechanism is worthy of inclusion and have added it to the manuscript, lines 231-235. While we agree that this possibility should be included, a couple of details inspire reflection: i) while illite is a 2:1 clay, unlike smectites, it generally does not possess interlayer hydration and swelling to accept water or much solutes, probably particularly anionic compounds like PFAS; ii) correlations of illite are stronger with the relatively long C5 and C6 compared to the smaller C3 and C4; iii) correlations for each PFAS with illite are less than the electrostatic charge of HAO, which has no interlayer spacing; and iv) for HAO, gibbsite and kaolinite, correlations with most PFAS are greater with electrostatic charge than with mineral abundance, bolstering electrostatics as an important component of sorption.

Reviewer response: Thanks

Figure 1: Panel C could be better utilized. For example, A and B show details of composition that support the caption, but there is nothing in panel C that really represents what is described in the caption for that panel. I.e., the panel doesn't really reflect anything about, "Shallow-aquifer flow zones tend to weather less extensively than the vadose zone; in reducing waters, Fe (hydr)oxides may undergo reductive dissolution thereby not contributing to the potential exchange complex, but Al (hydr)oxides and kaolinite can sorb short-chain PFAS regardless of oxidation state."

We thank the Reviewer for this comment. Figure 1 legend has been modified. The legend now reads:

Figure 1. Anthropogenic compounds applied to soils, e.g., via biosolids, manure, treated wastewater, aerial discharge, potentially can be accumulated by vegetation or soil organisms, or leach to underlying aquifers, where they partition between soil solids, water and, for volatiles, air-filled pore space. (a) Surface soils concentrate in weathering-resistant quartz, organic matter, and low concentrations of incipient amorphous Fe and Al (hydr)-oxides where long-chain PFAS are retained. (b) Subsurface horizons accumulate authigenic minerals, including clays and crystalline Fe and Al (hydr)-oxides where short-chain PFAS can sorb. (c) In saturated media, mineral assemblage commonly reflects more extensive weathering in high-flow zones relative to low-flow zones. Al (hydr)oxides and kaolinite can sorb short-chain PFAS regardless of oxidation state, while in reduced settings Fe (hydr)oxides may be subject to reductive dissolution thereby not contributing to the potential exchange complex.

Reviewer response: Thanks

Figure 3. There is an indication of a kaolinite line in Figure 3, panel C, but the kaolinite line is not visible. Thank you for this observation. Actually, the kaolinite line is present, hugging the right-hand border of the figure. Despite being a minor calculated (with the best available experimental data) contribution to positive bulk-soil charge, we include kaolinite because: i) kaolinite is a formal contributor on a bulk-soil basis based on recent AFM studies (Supplementary Information references 30 and 32); and ii) perhaps more importantly, whereas SI references 30 and 32 provide the best surface-site density values we have found, the value of surface-site density that they report is suspiciously low. We say this because the Al octahedral layer of kaolinite is iso-structural with gibbsite, yet the reported kaolinite surface-site density reported in these references is more than 100-fold lower than the value for gibbsite (see our Supplementary Table 5). Experimentally determining the surface site density on kaolinite appears to be quite challenging, and we do not want to criticize this pioneering work. However, we suspect that future studies reporting surface-site density for kaolinite might report higher values of surface site density. Such a possible future increase might well increase the kaolinite field substantially for Figures like our Figure 4 (formerly Figure 3).

Reviewer response: Thanks

Additional line-by-line comments for revision

Line 52. I suggest changing "biosolids-applied agricultural fields" to "biosolids-amended agricultural fields."

Line 69 should be edited to state, "...and it is well established..."

Line 239. It isn't clear how concentrations in excess of drinking water advisories informs slow degradation of side-chain polymers. Perhaps the authors intended to make a point about persistent concentrations that pose a health risk, but if so, this point is never made.

Line 284. Should this be "number of?"

Line 286. Should this be "implications for?"

Line 328. "So, the subsequent fate of PFAS in the subsurface, e.g., uptake by deeply rooted plants and/or leaching to groundwater resources, remains poorly predicted by these surface-soil data." I feel that the authors could be overstating things here. There are many reasons why leaching of PFAS to groundwater resources could be poorly predicted. Is it really the lack of info on interactions with the mineral phase? If so, is there data to support this? Since the current study is not applied in a predictive capacity (i.e., to model transport), it isn't clear if this knowledge gap is addressed by the current study. Figure 3 panel f. It seems like it would be more intuitive to plot Cs/Cw so that the plot can also give a sense for Kd without having to reverse the depicted trends during figure interpretation. Also, I suggest marking the eluviated horizons, depth typical of authigenic minerals, etc. on Figure 3 (at least on Panel A).

Figure 3 panel g. Please considering using the MCL of 4 ng/L as opposed to the interim advisory of 0.004 ng/L.

Figure 5 states "Heat map of nominal correlation coefficients for geochemical properties (columns), grouped according to close geochemical relationships, to PFCAs..." I assume you mean PFCA concentrations? It would be better to specify.

Same comment for PFASs.

Supplementary Figures 2 and 3 would benefit from a legend that defines what I is represented by each color. Supplementary Figure 11. It is impossible to correlate the colors in the graphs to the individual PFAS for PFCAs, PFASs, or other PFAS. Please consider use of patterning or colors that can be distinguished from one another enough to be matched to the legend.

Reviewer #3

(Remarks to the Author)

As mentioned in the previous round of my review, the detailed PFAS analysis coupled to extensive geological and geochemical information are of high interest and value. The authors well addressed all of my minor comments and I look forward to seeing future work towards a more generalizable translation of their findings.

REVIEWER COMMENTS

Reviewer #1 (Remarks to the Author):

The manuscript titled "Controls on the Fate of PFAS and Anthropogenic Anions in the Terrestrial Subsurface" presents a detailed examination of the mechanisms governing the mobility and fate of PFAS in soil and aquifer environments. It highlights the significant role of mineralogy, particularly authigenic minerals, in adsorbing PFAS, thereby influencing their environmental behavior and distribution.

Strengths:

The study's use of a wide array of geochemical and mineralogical analyses across different soil layers enhances the depth and reliability of the findings. This robust methodological framework provides a solid basis for the conclusions drawn.

The application of advanced geochemical modeling to predict PFAS interactions with soil minerals adds a valuable dimension to the research, offering insights that are both predictive and explanatory.

The detailed discussion on the impact of specific minerals like kaolinite and gibbsite on PFAS sorption is a strong point, providing clarity on the molecular interactions that govern PFAS mobility in subsurface environments.

The manuscript effectively uses data to challenge and refine existing theories on PFAS behavior, particularly concerning their interactions with different soil minerals.

We thank the Reviewer for the positive comments on the strengths of the manuscript.

Weaknesses:

While the data is generally well presented, some figures and tables are densely packed with information. Simplifying these or providing supplementary online material could help maintain the reader's focus on key results.

We agree that with so much data on the characterization of these sites, the figures and tables are dense. Figure 1 has been modified and split into two figures (now Figure 1 and Figure 2) for simplification. We have modified the text for broader audiences and information provided in the manuscript has been condensed to highlight key findings, with the majority of the data provided and expanded on in the 70-page Supplementary Information; for example, Figure 5 (previously Figure 4) of the manuscript is a condensed heat map of correlations found in the Supplementary Figures 22-24.

The manuscript could enhance its scope by incorporating more discussion on biological factors affecting PFAS fate, such as microbial degradation of precursors, which are equally crucial in subsurface environments.

We agree on the role of biological and microbe-facilitated transformations of PFAS precursors which occur in soils, including the side-chain fluorotelomer polymers (FTPs) typically applied to textiles. We have previously reported on transformation of FTPs, followed by PFAS intermediates, then recalcitrant PFAS – these papers are references 19 and 20 in this submitted manuscript. In this manuscript, we

summarized the FTP transformations we reported in our earlier papers, in lines 59-63. Due to word limitation, discussion on microbial degradation is expanded on in the Supplementary Information, section IV Extended PFAS Summary.

While the study presents detailed data, the generalization of results across varied environmental conditions could be better addressed to clarify the scope of application of the findings.

The heavy use of technical jargon and complex descriptions may make the paper less accessible to a broader audience for Nature Communications. Please include a nomenclature section to aid understanding.

We thank the Reviewer for this comment and we have edited the paper extensively for a broader audience. We now reserve more technical descriptions and details in the Supplementary Information (sections III Extended Geochemical Summary and IV Extended PFAS Summary) for researchers who wish to delve more deeply.

The manuscript's references are up-to-date; however, incorporating discussions on 'older' but relevant studies, such as Xiao et al., 2015, on PFAS in surface and subsurface soils could enhance its thoroughness.

We thank the Reviewer for the recommendation on use of this great reference and agree on the use of 'older' and relevant studies. We have included this reference in the main manuscript as well as several other references. Because of restrictions on both word count and number of references, further discussion with additional references from current to the 1960s can be found in the Supplementary Information.

Although the study provides a comprehensive current assessment, it could offer more explicit suggestions for future research directions, particularly in addressing the gaps identified in PFAS behavior under varying environmental conditions.

Thank you for this good point. In the last paragraph, we had suggested sorption experiments with the authigenic soil minerals we report as important. And based on this suggestion, we also added a suggestion to perform K_d experiments on the A and B horizon soils of major soil orders, e.g., entisols, alfisols, ultisols.

Reviewer #2 (Remarks to the Author):

General comments:

1. This dataset is significant and of high quality. I also think that it's inclusion of impacts from the textiles industry (not a major focus of the paper) is quite unique, and I wonder if there are aspects of the release type (textiles, side chains, etc.) that have yet to be explored in the manuscript. I think there's a lot that could be done with these data, and I'm excited to see where the authors take the story based on reviewer feedback.

This observation regarding textile industry is quite important and we plan to explore it more directly in future papers. While there are several other textile-PFAS papers, we are not aware of any with the

amount of data we have reported, nor the focus on textile sludge, which we believe other mills have spread on land as well. We are continuing research on wastes from this textile mill at present.

2. The significance of this study relative to prior work really needs to be clarified. As currently written, it is difficult to ascertain what this study contributes over prior work. Is it a conceptual model of PFAS sorption in the vadose zone and saturated zones with depth? Is it confirmation of a conceptual model that already existed with the extensive dataset the authors are presenting? There are a number of prior studies focused on PFAS sorption to minerals. The authors cite some of it (mostly in the supplementary material), but in my view the outcomes of this prior work were not thoroughly considered and were not used to identify a clear knowledge gap that is addressed here. It would be helpful if the authors outlined a clear conceptual model of the role of minerals in PFAS sorption based on outcomes of prior work, identified clearly knowledge gaps, and then provided clear hypotheses or objectives for the current study.

Thank you for this important point. Having worked intensely on this for so long, we did not step back far enough to offer perspective. We have edited the abstract and added a paragraph to the introductory section, lines 65-73 of the track-changes manuscript in simple mark-up mode, to position our study in context of current practice. In this paragraph, we reference a highly cited Feature Article in ES&T, identifying pH-dependent charge as important, but not reporting a quantitative method to calculate this property, a limitation that our submission addresses. In addition, for our tabulated summaries of the USDA soil database (Table 1, Supplementary Tables 19-21), we have added total organic carbon (TOC) to illustrate how TOC is greatly diminished in subsurface horizons compared to the surface, thereby limiting its role in PFAS fate in the subsurface relative to the organic-rich surface soils.

In case they are helpful, here are a few additional papers related to PFAS interactions with minerals that I'm not sure were considered:

- <https://doi.org/10.1016/j.chemosphere.2016.06.016>
- <https://doi.org/10.1021/acs.est.0c01646>
- <https://doi.org/10.1021/acs.est.2c01054>

We thank the Reviewer for these great reference suggestions.

Reference <https://doi.org/10.1021/acs.est.0c01646> was previously considered and is cited in the manuscript. The other references have been added to the manuscript at Line 245-247 (simple markup): In early research on PFAS interactions with minerals, Helsing et al. studied mineral sorption of four PFAS to alumina (positive surface charge) and silica (negative surface charge) crystals and observed sorption to Al_2O_3 but not SiO_2 , suggesting a critical role of electrostatic interactions⁵⁰. And have been added to the Supplementary Information at Lines 1019-1025: Interestingly, simulations found significant variation in molecular clustering and coordination between the PFAS and the hydroxyl surfaces of kaolinite among molecules of different chain-length and functional group, suggesting short-chain sulfonate PFAS molecules experience more restrictive transport in environmental soils with a higher kaolinite compared to carboxylates⁹⁷. These studies could provide some mechanistic insight for the significantly higher correlations for short-chain sulfonates with kaolinite in Field 1 unsaturated media relative to short-chain carboxylates (Supplementary Fig. 23).

3. Another aspect of significance that needs to be clarified is the importance of PFAS interactions with minerals in governing the overall distribution of PFAS. Showing correlations of PFAS with various mineral components is useful and interesting, but it does not inform the magnitude of the interactions of PFAS with those mineral phases (e.g. the magnitude of equilibrium sorption coefficients [K_d] values). Studies have shown that equilibrium sorption coefficients of PFAS to mineral phases can be quite low, and overall, the interactions of short-chain PFAS with any phase are quite low. So will these mineral phases really lead to a meaningful change in PFAS transport via sorption? Could we achieve similar predictions of transport, if we just assumed that there was no sorption to these phases? What about the role of other processes like matrix diffusion into/out of clays? I suspect the latter would be much more important in influencing PFAS transport over sorption to the minerals.

Thank you for these thoughts. The K_d values for shorter PFAS with mineral phases indeed can be modest.

Regarding the mineral phases, a central point we make in our manuscript, is that numerous common soil minerals have pH- and ionic-strength dependent surface charges (see our new Figure 4, formerly Figure 3). To represent potential sorption capacity for these minerals under real-world conditions, sorption experiments ideally should be carried out at pHs and ionic strengths similar to that in the subsurface, commonly with $\text{pH} < 5$ and ionic strength (I) on the order of 0.05. Commonly laboratory experiments are carried out at higher pH values (closer to the minerals' zero point of charge) and lesser ionic strengths, potentially leading to understated values of K_d .

Regarding the PFAS compounds, as the reviewer knows, sorption varies as a sensitive function of chain length. This is evident in our Figure 3 (Figure 2 in our original submission), Panel A, which depicts a sharp drop-off in ΣPFAS to a minimum at $\sim 1\text{m}$, near the base of eluviated horizons, then an increase to the subsurface maximum at 1.5 to 2 m, at depths typical of illuviated accumulation of authigenic minerals. This pattern also is reflected in Figure 3, Panel B, in that C5 through C7 also clearly exhibit local maxima at this depth interval. These details would not be predicted by a conceptual model of no sorption. In contrast, C4 shows no such maximum and, for these data, might be consistent with your question of whether assuming no sorption would be an effective conceptual model. Detailed knowledge of these behaviors are not well developed in natural systems and contributions like we report here are necessary for developing this knowledge.

On a larger-scale, even minor partitioning to mineral phases in the subsurface is important to understand because minor sorption, perhaps like C4, expressed over long flow-paths can lead to considerable accumulation. Such minor accumulation over longer flow-paths is important to appreciate in efforts to understand, for example: i) migration times of plumes, ii) magnitude of clean-up efforts, e.g., how long to expect pump-and-treat, and iii) potentially how to foster faster remediation, e.g., increasing pH to the ZPC, thereby diminishing sorption to the stationary phases.

Regarding diffusion into clays, we have added this as a possible mechanism in lines 231-235. While we definitely agree this is a potential mechanism of retention, whether this is a primary mode of sorption seems questionable: i) the PFAS we report upon are anions and the mineral surfaces are positively charged, and electrostatic attraction is a well-established fundamental phenomenon; ii) the perfluorinated chains of PFAS are mutually repulsive to and repulsed by higher dipole moments, and the hydrated diffuse interlayer of expanding smectite clays is well-ordered by the permanent electrostatic surface charge of smectite clays compared to the bulk aqueous phase, so the hydrophobic

PFAS chains might be more repulsed by the electrostatically ordered diffuse layer adjacent to clay surfaces than in the bulk water; iii) none of the minerals we report upon here are expanding smectite clays, so interlayer spacing is quite restricted; iv) correlations in Figure 5 (previously Figure 4) for the large majority of PFAS-mineral pairs are more significant with electrostatic charge than with mineral concentration; and v) for the shorter chains, PFBA through PFOA, correlations mostly increase with increasing chain-length, and diffusion into clay interlayers, to the extent it is important, would seem likely to favor shorter chains due to steric hindrance of longer chains in tight interlayers and repulsion of longer chains in the highly electrostatically ordered interlayer.

4. This study is very geology-oriented, uses a lot of the associated terminology, and it seems written for a geology audience over the type of broad readership often targeted by a journal such as Nature Communications. It will be difficult for a broader scientific audience to understand, and currently the relevance of this topic to a broad scientific audience isn't clear. PFAS are contaminants that are of global concern, but as written this manuscript seems better suited to a journal more focused on hydrogeology or environmental engineering. Clarifying the significance (as noted above) might go a long way to address this, but I also think that the final paragraphs of the paper need to do more to address the significance to a multidisciplinary readership.

Thank you, this is a good point. We have made substantial changes throughout the manuscript, taking care to make the article more readable for a general audience. Further discussion and more technical descriptions are reserved for the Supplementary Information.

We also agree that PFAS are of global concern. And much of the global fate of PFAS is governed by processes in the subsurface. We hope with our modifications to the manuscript, we can bring to light processes important to the fate of PFAS globally, without overburdening the reader with geochemical jargon.

Below are just a few, additional line by line comments.

Lines 17-31 (the abstract) are all background and a brief summary of the study objective. I suggest including a description of the significant outcomes of the study. Due to word limitations this may necessitate reducing the amount of background.

We thank the Reviewer for this suggestion. We have reworded the abstract substantially to reduce background and include more detail on the study outcome.

Lines 19-30 now read:

Per- and polyfluoroalkyl substances (PFAS), like many anthropogenic compounds, migrate into the environment through various means, e.g., soil-amendment impurities and ambient atmospheric deposition, potentially resulting in vegetative uptake and migration to groundwater. Existing approaches for modeling sorption of PFAS and other compounds commonly include treating soil as an undifferentiated homogeneous medium, with distribution constants (e.g., K_d , K_{oc}) generated for individual compounds empirically using surface soils. Taking into consideration the limited mineral variety expected in weathered geologic media, mobility of PFAS and other anthropogenic compounds can be better understood by accounting for these predictable mineral assemblages. Here we explore

the role of minerals and electrostatic sorption in controlling PFAS mobility in subsurface soils and shallow aquifers at agricultural sites following heavy contamination by measuring geochemical parameters and PFAS, and calculating pH-dependent mineral surface charges through full soil and aquifer columns. Furthermore, we report on the ubiquitous distribution of these minerals in U.S. soils.

Line 42: Minor comment that “and the like” is a vague way to end the sentence. Plus this information is unsupported by references.

Thank you for the comment; we removed “and the like” from the sentence.

Lines 40-41 now reads: Often these biosolids bear a witches’ brew of our societal waste, e.g., pharmaceuticals, PFAS, and plasticizers ².

Line 50: Why only advection? Why not dispersion, back-diffusion, and other mechanisms?

Thank you for pointing that out; we have made the statement more general to include other groundwater solute transport mechanisms.

Lines 44-48 now reads: Released to the land’s surface, these pesticides, and manure-, biosolids- and atmospheric-borne chemicals interact intimately with soils, which initially act as sinks from the aqueous phase, serving as sizable reservoirs, and ultimately time-delayed sources for moderated concentrations of recalcitrant contaminants, perhaps to be accumulated in vegetation or soil organisms ⁶ or to leach through the vadose zone and migrating to groundwater ^{7,8}.

Line 64 mentions fluorotelomer polymers, and I think it would be beneficial to denote that these are also PFAS (i.e., polymeric PFAS) that can degrade to non-polymeric PFAS. As written it implies that these are polymers (not PFAS) that can degrade into PFAS.

We have added the additional information to make this clearer to the reader.

Lines 56-59 now reads: Among PFAS uses, the widespread industries of textile¹⁵, paper¹⁶, and carpet¹⁷ milling operations historically have used polymeric PFAS, e.g., sidechain fluorotelomer polymers, to treat their products for anti-staining properties ¹⁸.

Lines 78-80: Please consider revising as this is not a complete sentence and it is unclear what the authors are trying to say.

We have modified this sentence for clarity and grammar.

Lines 80-83 now reads: Here we report PFAS distribution patterns through the subsurface, document the ubiquitous distribution of authigenic minerals in soils across the United States, and explore how these globally distributed minerals and other geologic soil parameters control distribution and fate of PFAS in the subsurface (SI II).

Lines 117-135: Some of this information was already mentioned in the introduction, and this is just more general background on PFAS. If this space is to be used to focus on PFAS background, then I suggest using to define why this study is needed. So far the authors have simply said that current work focuses on surface soils, which are higher in Natural Organic Matter (NOM) than deeper soils. However, they haven’t said anything about why understanding interactions with minerals in more weathered depths is important or what kinds of interactions are not already understood via prior work.

We thank the reviewers for drawing our attention to this section. We removed this entire section from the manuscript; information which has already been mentioned was removed and new details were moved to the introduction and following section. We have added a paragraph to the introductory section, lines 65-73, to make clear that the subsurface is composed of minerals, that contaminant-mineral interaction is conceptually understood to be important in subsurface fate, but that specific minerals in real-world soil bodies, and quantitative expression of mineral properties that might affect fate, are not as well defined.

Line 240 discusses the strong correlation between negatively charged illite and concentrations of short-chain (and anionic) PFAS. Multiple hypotheses are given regarding the cause, but none address non-sorptive mechanisms such as the diffusion of PFAS into clay, which would presumably occur faster for short-chain PFAS due to low molecular weight.

We agree that this potential mechanism is worthy of inclusion and have added it to the manuscript, lines 231-235.

While we agree that this possibility should be included, a couple of details inspire reflection: i) while illite is a 2:1 clay, unlike smectites, it generally does not possess interlayer hydration and swelling to accept water or much solutes, probably particularly anionic compounds like PFAS; ii) correlations of illite are stronger with the relatively long C5 and C6 compared to the smaller C3 and C4; iii) correlations for each PFAS with illite are less than the electrostatic charge of HAO, which has no interlayer spacing; and iv) for HAO, gibbsite and kaolinite, correlations with most PFAS are greater with electrostatic charge than with mineral abundance, bolstering electrostatics as an important component of sorption.

Figure 1: Panel C could be better utilized. For example, A and B show details of composition that support the caption, but there is nothing in panel C that really represents what is described in the caption for that panel. I.e., the panel doesn't really reflect anything about, "Shallow-aquifer flow zones tend to weather less extensively than the vadose zone; in reducing waters, Fe (hydr)oxides may undergo reductive dissolution thereby not contributing to the potential exchange complex, but Al (hydr)oxides and kaolinite can sorb short-chain PFAS regardless of oxidation state."

We thank the Reviewer for this comment. Figure 1 legend has been modified. The legend now reads:

Figure 1. Anthropogenic compounds applied to soils, e.g., via biosolids, manure, treated wastewater, aerial discharge, potentially can be accumulated by vegetation or soil organisms, or leach to underlying aquifers, where they partition between soil solids, water and, for volatiles, air-filled pore space. (a) Surface soils concentrate in weathering-resistant quartz, organic matter, and low concentrations of incipient amorphous Fe and Al (hydr)-oxides where long-chain PFAS are retained. (b) Subsurface horizons accumulate authigenic minerals, including clays and crystalline Fe and Al (hydr)-oxides where short-chain PFAS can sorb. (c) In saturated media, mineral assemblage commonly reflects more extensive weathering in high-flow zones relative to low-flow zones. Al (hydr)oxides and kaolinite can sorb short-chain PFAS regardless of oxidation state, while in reduced settings Fe (hydr)oxides may be subject to reductive dissolution thereby not contributing to the potential exchange complex.

Figure 3. There is an indication of a kaolinite line in Figure 3, panel C, but the kaolinite line is not visible.

Thank you for this observation. Actually, the kaolinite line is present, hugging the right-hand border of the figure.

Despite being a minor *calculated* (with the best available experimental data) contribution to positive bulk-soil charge, we include kaolinite because: i) kaolinite is a formal contributor on a bulk-soil basis based on recent AFM studies (Supplementary Information references 30 and 32); and ii) perhaps more importantly, whereas SI references 30 and 32 provide the best surface-site density values we have found, the value of surface-site density that they report is suspiciously low. We say this because the Al octahedral layer of kaolinite is iso-structural with gibbsite, yet the reported kaolinite surface-site density reported in these references is more than 100-fold lower than the value for gibbsite (see our Supplementary Table 5). Experimentally determining the surface site density on kaolinite appears to be quite challenging, and we do not want to criticize this pioneering work. However, we suspect that future studies reporting surface-site density for kaolinite might report higher values of surface site density. Such a possible future increase might well increase the kaolinite field substantially for Figures like our Figure 4 (formerly Figure 3).

Reviewer #3 (Remarks to the Author):

The authors present an interesting dataset combining field data from a large number of soil cores (~50/site) with extensive geological and geochemical information complemented with detailed PFAS analysis. These data and explaining the interesting trends in CF₂ chain length dependent distribution, especially in deeper, carbon depleted soil and aquifer layers, form the core of the manuscript, which is very interesting and valuable on its own.

However, the authors could have substantially increased the value of their work, had they further validated their interpretations with laboratory scale experiments (e.g. PFCAs formation rates from precursors as well as sorption and desorption batch or column experiments at relevant conditions as partially suggested by the authors in line 296ff). This would have helped complementing their field data with some sort of kinetic modeling of PFCAs fate (formation, transport sorption in dependence of water saturation, soil/aquifer composition, AEC, CEC, pH, ionic strength etc.), to better contextualize, translate and enable extrapolation of the interesting findings shown in Figure 4. Such complementary activities would have given the study more weight and could have facilitated the translation of these findings to other sites and scenarios.

As the manuscript stands now, it is hard to say if the trends and findings on chain length dependent behavior and the importance of electrostatic interactions in carbon depleted matrices are a “just” really interesting observation from two field sites, or if -and how- these findings can be generalized and used for future modeling, risk assessment, and potentially prioritization of mitigation measures across the large number of potentially relevant sites mentioned by the authors in line.

We thank the Reviewer for this suggestion. Our central goal was to report real-world PFAS patterns and how they relate to environmental properties following a large-scale contamination and document that soil properties we show to be impacting PFAS fate at our study sites are ubiquitously distributed in soils across the United States.

We agree that laboratory experiments with PFAS precursor transformations are useful. In fact, our laboratory has conducted these experiments with commercial side-chain fluorotelomer polymers (FTPs), the polymeric compounds most commonly used on textiles; we reference these papers a couple of times in our manuscript, references 19 and 20 in the manuscript. In these papers (and Supplementary Information), we report the rates of environmental transformation of FTPs and all PFAS intermediates leading to terminal PFAS. We agree that additional experiments are a logical and important next step for this project and are planning additional studies based on our findings. However, due to the sheer quantity of analyses and modeling we report here, the scale of the in-depth characterization at these sites including exhaustive measurements and calculations, and the cumulative ~90 pages of data and interpretations we already included in the manuscript and Supplementary Information, we believe that further laboratory studies are beyond the scope of this already large-scale study and better suited for future papers as we suggest in the manuscript.

Minor remarks:

Line 52-56: a bit of an overstatement on the lack of studies on minerals and NOM, as these aspects have been the focus of a substantial number of studies over the last decade(s), including for charged and ionizable compounds.

We thank the Reviewer for the comment. Our point was that environmental studies on subsurface PFAS that also describe minerals and NOM, particularly at depths of the ~1 to 20 meter interval we report upon are relatively rare. We have removed this statement to avoid overstating and oversimplification.

Line 62: a sentence contextualizing the focus on sidechain fluorotelomer within the larger family of PFAS may be useful for some readers.

Due to limitations on word count, much of the discussion has been relegated to the Supplementary Information (section IV Extended PFAS Summary).

More detail has been added to the manuscript, referencing our earlier papers on sidechain fluorotelomer polymers, at lines 59-64 in simple markup mode:

These polymers slowly degrade by abiotic hydrolysis, with half-lives on the order of 55 to 90 years under environmental conditions, to form fluorotelomer alcohols (n :2FTOHs – $F(CF_2)_n(CH_2)_2OH$, $n \sim 4$ -18 even integers) as first-generation products¹⁹, and ultimately forming recalcitrant perfluorocarboxylates (PFCAs – $F(CF_2)_nCO(OH)$, $n \sim 3$ -18) and related compounds in the environment by microbially mediated transformations^{19,20}. Along with recalcitrant perfluorosulfonates (PFSAs – $F(CF_2)_nSO_3H$, $n \sim 4$ -12), PFCAs are among the most widely distributed PFAS in the environment¹³.

Line 73: rephrase “in soils in our soils and soils”

Thank you for catching this oversight. We have reworded this sentence to read: Here we report PFAS distribution patterns through the subsurface, document the ubiquitous distribution of authigenic minerals in soils across the United States, and explore how these globally distributed minerals and other geologic soil parameters control distribution and fate of PFAS in the subsurface (SI II).

Line 219: this paragraph could be extended somewhat.

We used this paragraph introducing the overarching findings of the relationships, and further discussion on each of the individual points can be found in subsequent paragraphs. Instead of breaking this up into separate parts, we have combined this discussion into a single paragraph for clarity.

Line 261: some systematic experiments would have been possible and highly appreciated to get a better understating on the sorption processes. Please also see my general comments.

We thank the reviewers for this comment and agree that laboratory scale studies are important. However, these would be outside of the scope of this study, which is a focus on a thorough characterization of two real-world datasets, including PFAS and mineral assessment of two sites and mineral assessment of a larger, national-wide database. This study can inform and guide selection of what future laboratory studies would be most impactful. Our data- and modeling-dense >70-page Supplementary Information offers many advances to the state of the science and speaks to the effort we have put forth in this study so far.

Responses to comments for Nature Communications manuscript # NCOMMS-24-57629-A

February 28, 2025

We thank all of the Reviewers for their thorough reviews, thoughtful edits and comments, and careful consideration of our manuscript. For clarity, new Reviewer comments intended for response are highlighted in green and new Author responses are in bold.

Reviewer #2 (Remarks to the Author):

I would like to thank the authors for simplifying the use of technical jargon, improvements to figures, and improved framing of the contribution of their work. As I understand it, important contributions of this work include use of calculated, pH-dependent charge when correlating subsurface PFAS concentrations with geochemical parameters, an improved conceptual model of PFAS interactions with subsurface soils that includes the relative importance of various minerals, and an evaluation of the global ubiquity of these minerals based on the United States Department of Agriculture (USDA) soils database. I agree that these results offer important considerations for those evaluating PFAS fate and transport either through laboratory experiments or modeling. However, **I think it is the application of these results that will be of most interest to a broad, multidisciplinary audience consistent with Nature Communications. For example, it would be great to see a quantitative analysis (e.g., fate and transport model- could be a simplified one) that demonstrates how consideration of these mineral-specific interactions influence the distribution of short-chain PFAS vs. a model that considers homogenous surface soil composition vs. (perhaps) a model that considers little to no sorption to minerals. Without that application, it is difficult to frame the importance of the authors' advancements to the PFAS sorption conceptual model.** I have provided some point by point feedback within my original comments and the authors' responses in the document below. There are some additional, minor comments unique to this version beneath this point by point response. All line numbers reference the track changes version of the manuscript.

Coauthor response: We agree that additional experimentation and further application and modeling would be of broad interest to the readers and hope that it will be the subject of future work from our lab. However, we currently do not have the required experimental data to adequately model the recommended approach, i.e. the intrinsic sorption constants for each solid-ion pair for the reported minerals. The only available values are the surface soil K_d values, which we argue are not valid in the subsurface environment, nor the accompanying variations in pH and ionic strength. Without these required experimental values, additional modeling is outside the scope of our intended study.

2. The significance of this study relative to prior work really needs to be clarified. As currently written, it is difficult to ascertain what this study contributes over prior work. Is it a conceptual model of PFAS sorption in the vadose zone and saturated zones with depth? Is it confirmation of a conceptual model that already existed with the extensive dataset the authors are presenting? There are a number of prior studies focused on PFAS sorption to minerals. The authors cite some of it (mostly in the supplementary

material), but in my view the outcomes of this prior work were not thoroughly considered and were not used to identify a clear knowledge gap that is addressed here. It would be helpful if the authors outlined a clear conceptual model of the role of minerals in PFAS sorption based on outcomes of prior work, identified clearly knowledge gaps, and then provided clear hypotheses or objectives for the current study.

Thank you for this important point. Having worked intensely on this for so long, we did not step back far enough to offer perspective. We have edited the abstract and added a paragraph to the introductory section, lines 65-73 of the track-changes manuscript in simple mark-up mode, to position our study in context of current practice. In this paragraph, we reference a highly cited Feature Article in ES&T, identifying pH-dependent charge as important, but not reporting a quantitative method to calculate this property, a limitation that our submission addresses. In addition, for our tabulated summaries of the USDA soil database (Table 1, Supplementary Tables 19-21), we have added total organic carbon (TOC) to illustrate how TOC is greatly diminished in subsurface horizons compared to the surface, thereby limiting its role in PFAS fate in the subsurface relative to the organic-rich surface soils.

Reviewer response: Lines 65-73 of the track changes manuscript do not appear to reflect any tracked changes, and do not appear to discuss the topics mentioned in the authors' response (e.g., pH-dependent charge).

In the tracked changes manuscript, with Simple Markup visible, the paragraph at lines 65-73 was added and is new to offer the perspective mentioned. This is the paragraph starting with "Despite this considerable potential..." to "... are less well explored." (lines 68-77 of the current version). In this paragraph, we reference the ES&T Feature article by Sigmund et al. (manuscript citation 22).

3. Another aspect of significance that needs to be clarified is the importance of PFAS interactions with minerals in governing the overall distribution of PFAS. Showing correlations of PFAS with various mineral components is useful and interesting, but it does not inform the magnitude of the interactions of PFAS with those mineral phases (e.g. the magnitude of equilibrium sorption coefficients [Kd] values). Studies have shown that equilibrium sorption coefficients of PFAS to mineral phases can be quite low, and overall, the interactions of short-chain PFAS with any phase are quite low. So will these mineral phases really lead to a meaningful change in PFAS transport via sorption? Could we achieve similar predictions of transport, if we just assumed that there was no sorption to these phases? What about the role of other processes like matrix diffusion into/out of clays? I suspect the latter would be much more important in influencing PFAS transport over sorption to the minerals.

Thank you for these thoughts. The Kd values for shorter PFAS with mineral phases indeed can be modest.

Regarding the mineral phases, a central point we make in our manuscript, is that numerous common soil minerals have pH- and ionic-strength dependent surface charges (see our new Figure 4, formerly Figure 3). To represent potential sorption capacity for these minerals under real-world conditions, sorption

experiments ideally should be carried out at pHs and ionic strengths similar to that in the subsurface, commonly with $\text{pH} < 5$ and ionic strength (I) on the order of 0.05. Commonly laboratory experiments are carried out at higher pH values (closer to the minerals' zero point of charge) and lesser ionic strengths, potentially leading to understated values of K_d .

Regarding the PFAS compounds, as the reviewer knows, sorption varies as a sensitive function of chain length. This is evident in our Figure 3 (Figure 2 in our original submission), Panel A, which depicts a sharp drop-off in ΣPFAS to a minimum at $\sim 1\text{m}$, near the base of eluviated horizons, then an increase to the subsurface maximum at 1.5 to 2 m, at depths typical of illuviated accumulation of authigenic minerals. This pattern also is reflected in Figure 3, Panel B, in that C5 through C7 also clearly exhibit local maxima at this depth interval. These details would not be predicted by a conceptual model of no sorption. In contrast, C4 shows no such maximum and, for these data, might be consistent with your question of whether assuming no sorption would be an effective conceptual model. Detailed knowledge of these behaviors are not well developed in natural systems and contributions like we report here are necessary for developing this knowledge.

On a larger-scale, even minor partitioning to mineral phases in the subsurface is important to understand because minor sorption, perhaps like C4, expressed over long flow-paths can lead to considerable accumulation. Such minor accumulation over longer flow-paths is important to appreciate in efforts to understand, for example: i) migration times of plumes, ii) magnitude of clean-up efforts, e.g., how long to expect pump-and-treat, and iii) potentially how to foster faster remediation, e.g., increasing pH to the ZPC, thereby diminishing sorption to the stationary phases.

Regarding diffusion into clays, we have added this as a possible mechanism in lines 231-235. While we definitely agree this is a potential mechanism of retention, whether this is a primary mode of sorption seems questionable: i) the PFAS we report upon are anions and the mineral surfaces are positively charged, and electrostatic attraction is a well-established fundamental phenomenon; ii) the perfluorinated chains of PFAS are mutually repulsive to and repulsed by higher dipole moments, and the hydrated diffuse interlayer of expanding smectite clays is well-ordered by the permanent electrostatic surface charge of smectite clays compared to the bulk aqueous phase, so the hydrophobic PFAS chains might be more repulsed by the electrostatically ordered diffuse layer adjacent to clay surfaces than in the bulk water; iii) none of the minerals we report upon here are expanding smectite clays, so interlayer spacing is quite restricted; iv) correlations in Figure 5 (previously Figure 4) for the large majority of PFAS-mineral pairs are more significant with electrostatic charge than with mineral concentration; and v) for the shorter chains, PFBA through PFOA, correlations mostly increase with increasing chain-length, and diffusion into clay interlayers, to the extent it is important, would seem likely to favor shorter chains due to steric hindrance of longer chains in tight interlayers and repulsion of longer chains in the highly electrostatically ordered interlayer.

Reviewer response: The authors indicate that minor sorption can lead to considerable accumulation, but this is purely conceptual, and I think that constitutes a remaining weakness of the manuscript. The authors have not done any quantitative analysis to demonstrate the degree to which these mineral interactions could influence fate and transport. It seems like it would be feasible to use site conditions and simplifying assumptions to model how transport looks with and without these charge-based interactions. It doesn't necessarily need to be a model that is fitted to field data, just application of the study outcomes in a more quantitative way that would demonstrate, for example, migration of PFBA over a 20 year period with and without these mineral interactions.

Coauthor response: Unfortunately, the data required to perform quantitative modeling simply do not exist yet. While our work provides the theoretical foundation for the electrostatic component of sorption, rigorous modeling requires that the electrostatic component of sorption be superimposed on the intrinsic chemical sorption constant (i.e., K^{int} in Supplementary Equations 1 through 16). We make this very point, that the manuscript reports methodology to quantify the electrostatic component of sorption but the chemical component requires future investigation, in lines 258-260 where we state, *“Unfortunately, unlike the chemically indifferent forces of the electrostatic interactions explored here, the intrinsic component of sorption is ion and surface specific, and therefore remains a challenge for future investigations”*.

We hope to report values of K^{int} in the future, at which time we would proceed to explore the impact of subsurface sorption with quantitative models.

Below are just a few, additional line by line comments.

Lines 17-31 (the abstract) are all background and a brief summary of the study objective. I suggest including a description of the significant outcomes of the study. Due to word limitations this may necessitate reducing the amount of background.

We thank the Reviewer for this suggestion. We have reworded the abstract substantially to reduce background and include more detail on the study outcome.

Lines 19-30 now read:

Per- and polyfluoroalkyl substances (PFAS), like many anthropogenic compounds, migrate into the environment through various means, e.g., soil-amendment impurities and ambient atmospheric deposition, potentially resulting in vegetative uptake and migration to groundwater. Existing approaches for modeling sorption of PFAS and other compounds commonly include treating soil as an undifferentiated homogeneous medium, with distribution constants (e.g., K_d , K_{oc}) generated for individual compounds empirically using surface soils. Taking into consideration the limited mineral variety expected in weathered geologic media, mobility of PFAS and other anthropogenic compounds can be better understood by accounting for these predictable mineral assemblages. Here we explore the role of minerals and electrostatic sorption in controlling PFAS mobility in subsurface soils and shallow aquifers at agricultural sites following heavy contamination by measuring geochemical parameters and

PFAS, and calculating pH-dependent mineral surface charges through full soil and aquifer columns. Furthermore, we report on the ubiquitous distribution of these minerals in U.S. soils.

Reviewer response: Thank you for the edits to your abstract; however, I still do not see any mention of the primary study outcomes within the edited version. For example, “Here we report the role of minerals and electrostatic sorption in controlling PFAS mobility in the subsurface soils and shallow aquifers at agricultural sites following heavy contamination by measuring geochemical parameters and PFAS, calculating pH-dependent mineral surface charges through full soil and aquifer columns,” is a summary of approach. Not an outcome.

Coauthor response: We have added more detail to the abstract. The abstract now reads:

“Per- and polyfluoroalkyl substances (PFAS), like many anthropogenic compounds, migrate into the environment through various means, e.g., soil-amendment impurities and ambient atmospheric deposition, potentially resulting in vegetative uptake and migration to groundwater. Existing approaches for modeling sorption of PFAS and other compounds commonly include treating soil as an undifferentiated homogeneous medium, with distribution constants (e.g., K_d , K_{oc}) generated for individual compounds empirically using surface soils. Taking into consideration the limited mineral variety expected in weathered geologic media, mobility of PFAS and other anthropogenic compounds can be better understood by accounting for these predictable mineral assemblages. Here we explore the role of minerals and electrostatic sorption in controlling PFAS mobility in subsurface soils and shallow aquifers at agricultural sites following heavy contamination by measuring geochemical parameters and PFAS, and calculating pH-dependent mineral surface charges through full soil and aquifer columns. These data suggest the mobility of short-chain PFAS in the subsurface is controlled largely by aluminum-oxide mineral(oid)s, clay concentrations, and electrostatic charges, whereas long-chain PFAS mobility is controlled by organic matter and the air-water interfacial area. Furthermore, we report on the ubiquitous distribution of these minerals in U.S. soils.”

Additional line-by-line comments for revision

Line 52. I suggest changing “biosolids-applied agricultural fields” to “biosolids-amended agricultural fields.”

Coauthor response: Done, thank you.

Line 69 should be edited to state, “...and it is well established...”

Coauthor response: Done, thank you.

Line 239. It isn’t clear how concentrations in excess of drinking water advisories informs slow degradation of side-chain polymers. Perhaps the authors intended to make a point about persistent concentrations that pose a health risk, but if so, this point is never made.

Coauthor response: This comment references lines 195-201 of our resubmitted manuscript (simple markup mode). Our point here makes sense when it is read in its original context with the two preceding points. In context, we state the logic follows from three observations, which we delineate i through iii, when they are considered “in concert”.

We have added the detail that some of our detections are short-lived compounds, found sixteen years after sludge application. With this change, in full context of the paragraph, we think our point is well expressed.

Line 284. Should this be “number of?”

Coauthor response: Done, thank you.

Line 286. Should this be “implications for?”

Coauthor response: No, we do not believe that “offers implications both local and large-scale..” should read “offers implications for both local and large-scale..”

Line 328. “So, the subsequent fate of PFAS in the subsurface, e.g., uptake by deeply rooted plants and/or leaching to groundwater resources, remains poorly predicted by these surface-soil data.” I feel that the authors could be overstating things here. There are many reasons why leaching of PFAS to groundwater resources could be poorly predicted. Is it really the lack of info on interactions with the mineral phase? If so, is there data to support this? Since the current study is not applied in a predictive capacity (i.e., to model transport), it isn't clear if this knowledge gap is addressed by the current study.

Coauthor response: Thanks for this observation. This sentence was replaced in lines 284-285 of our resubmitted manuscript with this: “So the applicability of these surface-soil values for accurately modeling fate of PFAS in the subsurface is dubious”.

Figure 3 panel f. It seems like it would be more intuitive to plot C_s/C_w so that the plot can also give a sense for K_d without having to reverse the depicted trends during figure interpretation. Also, I suggest marking the eluviated horizons, depth typical of authigenic minerals, etc. on Figure 3 (at least on Panel A).

Coauthor response: We thank you for your recommendation. Our intention with panels f and g was to depict the state of the groundwater specifically. In panel f, the groundwater/surface soil ratio was plotted to illustrate the higher mobility of the shorter-chain PFAS through soils to groundwater relative to the longer-chains.

In continued pursuit of simplifying the figures, as suggested by the Reviewers, we think it is best to report the authigenic mineral distribution with depth in Figure 2b. For commonality of reference between the geochemistry of Figure 2 and the PFAS chemistry of Figure 3, in addition to the depth depicted on the y axis, the water table is represented in both Figure 2b and Figure 3a-b to guide the reader.

Figure 3 panel g. Please considering using the MCL of 4 ng/L as opposed to the interim advisory of 0.004 ng/L.

Coauthor response: Thank you for this recommendation, we have changed Figure 3 panel G levels to reflect this change.

Figure 5 states “Heat map of nominal correlation coefficients for geochemical properties (columns), grouped according to close geochemical relationships, to PFCAs...” I assume you mean PFCA concentrations? It would be better to specify. Same comment for PFSAs.

Coauthor response: Done, thank you.

Supplementary Figures 2 and 3 would benefit from a legend that defines what is represented by each color.

Coauthor response: Legends have been added to the figures (Supplementary Figures 2-4).

Supplementary Figure 11. It is impossible to correlate the colors in the graphs to the individual PFAS for PFCAs, PFSA, or other PFAS. Please consider use of patterning or colors that can be distinguished from one another enough to be matched to the legend.

Coauthor response: Done, thank you.